# THREE PROBLEM CLASSES THAT MARKOV REWARDS CANNOT EXPRESS

## ABSTRACT

In this paper, we study the expressivity of Markovian reward functions, and identify several limitations to what they can express. Specifically, we look at three classes of reinforcement learning tasks (multi-objective reinforcement learning, risk-averse reinforcement learning, and modal reinforcement learning), and then prove mathematically that most of the tasks in each of these classes cannot be expressed using scalar, Markovian reward functions. In the process, we provide necessary and sufficient conditions for when a multi-objective reinforcement learning problem can be reduced to ordinary, scalar reward reinforcement learning. We also call attention to a new class of reinforcement learning problems (namely those we call "modal" problems), which have so far not been given any systematic treatment in the reinforcement learning literature. In addition, we also show that many of these problems *can* be solved effectively using reinforcement learning. This rules out the possibility that those problems which cannot be expressed using Markovian reward functions also are impossible to learn effectively.

## 1 INTRODUCTION

To use reinforcement learning (RL) to solve a task, it is necessary to first encode that task using a reward function (Sutton & Barto, 2018). Usually, these reward functions are *Markovian* functions from *state-action-next-state* triples to reals. In this paper, we study the expressivity of Markovian reward functions, and identify several limitations to what they can express. Specifically, we will examine three classes of tasks, all of which are both intuitive to understand, and useful in practical situations. We will then show that *almost all* tasks in each of these three classes are impossible to express using Markovian reward functions. Moreover, we also show that many of these problems *can* be solved effectively with RL, either by providing references to existing literature, or by providing an outline of a possible approach. This rules out the possibility that those problems which cannot be expressed using Markovian reward functions also are impossible to learn effectively.

The first class of problems we look at, in Section 2, is the single-policy version of multi-objective RL (MORL). In such a problem, the agent receives multiple reward signals, and the aim is to learn a single policy that achieves an optimal trade-off of those rewards according to some criterion (Roijers et al., 2013; Liu et al., 2015). For example, a single-policy MORL algorithm might attempt to maximise the rewards lexicographically (Skalse et al., 2022b). We will look at the question of which MORL problems can be reduced to ordinary RL, by providing a scalar reward function that induces the same preferences as the original MORL problem. Moreover, we will provide a complete solution to this problem, in the form of necessary and sufficient conditions. We find that this can *only* be done for MORL problems that correspond to a linear weighting of the rewards, which means that it cannot be done for the vast majority of all interesting MORL problems.

The next class of problems we look at, in Section 3, is risks-sensitive RL. There are many contexts where it is desirable to be risk averse. In economics, and related fields, this is often modelled using utility functions $U : \mathbb{R} \to \mathbb{R}$ which are concave in some underlying quantity. Can the same thing be done with reward functions? Is it possible to take a reward function, and then create a version of that reward function which induces more risk-averse behaviour? We show that the answer is no – none of the standard risk-averse utility functions can be expressed using reward functions. This demonstrates another limitation in the expressive power of Markovian rewards.

The last class of problems we look at, in Section 4, is something we call *modal* tasks. These are tasks where the agent is evaluated not only based on what trajectories it generates, but also based on what it *could have done* along those trajectories. For example, consider the instruction "you should always be *able* to return to the start state". We provide a formalisation of such tasks, argue that there are many situations in which these tasks could be useful, and finally prove that these tasks also typically cannot be formalised using ordinary reward functions.

In Section 5, we discuss how to solve tasks from each of these classes using RL. We provide references to existing literature, and then sketch both an approach for learning a wide class of MORL problems, and an approach for learning a wide class of modal problems. Finally, in Section 6, we discuss the significance and limitations of our results, together with ways to extend them.

## 1.1 RELATED WORK

There has been a few recent papers which examine the expressivity of Markovian reward functions. The first of these is the work by Abel et al. (2021), who point to three different ways to formalise the notion of a "task" (namely, as a set of acceptable policies, as an ordering over policies, or as an ordering over trajectories). They then demonstrate that each of these classes contains at least one instance which cannot be expressed using a reward function (by using the fact that the set of all optimal policies forms a convex set, and the fact that the reward function is Markovian). They also provide algorithms which compute reward functions for these types of tasks, by constructing a linear program. We greatly extend their work by providing new results that are significantly stronger.

Another important paper is the work by Vamplew et al. (2022), who argue that there are many important aspects of intelligence which can be captured by MORL, but not by scalar RL. Like them, we also argue that MORL is a genuine extension of scalar RL, but our approach is quite different. They focus on the question of whether MORL or (scalar) RL is a better foundation for the development of general intelligence (considering feasibility, safety, and etc), and they provide qualitative arguments and biological evidence. By contrast, we are more narrowly focused on what incentive structures can be expressed by MORL and scalar RL, and our results are mathematical.

There is also other relevant work that is less strongly related. For example, Icarte et al. (2022) point out that there are certain tasks which cannot be expressed using Markovian rewards, and propose a way extend their expressivity by augmenting the reward function with an automaton that they call a *reward machine*. Similar approaches have also been used by e.g. Hasanbeig et al. (2020); Hammond et al. (2021). There are also other ways to extend Markovian rewards to a more general setting, such as *convex RL*, as studied by e.g. Hazan et al. (2019); Zhang et al. (2020); Zahavy et al. (2021); Geist et al. (2022); Mutti et al. (2022), and *vectorial RL*, as studied by e.g. Cheung (2019a;b). Also related is the work by Skalse et al. (2022c), who show that there are certain relationships that are never satisfied by any pair of reward functions. This paper can also be seen as relating to earlier work on characterising what kinds of preference structures can be expressed using utility functions, such as the famous work by von Neumann & Morgenstern (1947), and other work in game theory.

There is a large literature on (the overlapping topics of) single-policy MORL, constrained RL, and risk-sensitive RL. Some notable examples of this work includes Achiam et al. (2017); Chow et al. (2017); Miryoosefi et al. (2019); Tessler et al. (2019); Skalse et al. (2022b). This existing literature typically focuses on the creation of algorithms for solving particular MORL problems, and has so far not tackled the problem of characterising when MORL problems can be reduced to scalar RL. Modal RL has (to the best of our knowledge) never been discussed explicitly in the literature before. However, it relates to some existing work, such as side-effect avoidance (Krakovna et al., 2018; 2020; Turner et al., 2020), and the work by Wang et al. (2020).

## 1.2 PRELIMINARIES

The standard RL setting is formalised using *Markov Decision Processes* (MDPs), which are tuples $\langle \mathcal{S}, \mathcal{A}, \tau, \mu_0, \mathbb{R}, \gamma \rangle$ where $\mathcal{S}$ is a set of states, $\mathcal{A}$ is a set of actions, $\tau : \mathcal{S} \times \mathcal{A} \rightsquigarrow \mathcal{S}$ is a transition function, $\mu_0$ is an initial state distribution over $\mathcal{S}$, $R : \mathcal{S} \times \mathcal{A} \times \mathcal{S} \rightsquigarrow \mathbb{R}$ a reward function, where $R(s, a, s')$ is the reward obtained if the agent moves from state $s$ to $s'$ by taking action $a$, and $\gamma \in (0, 1)$ is a discount factor. Here, $f : X \rightsquigarrow Y$ denotes a probabilistic mapping $f$ from $X$ to $Y$. A state is *terminal* if $\tau(s, a) = s$ and $R(s, a, s) = 0$ for all $a$. A *trajectory* $\xi$ is a path $s_0, a_0, s_1 \ldots$ in an

MDP that is possible according to $\mu_0$ and $\tau$. We use $G$ to denote the *trajectory return function*, where $G(\xi) = \sum_{t=0}^{\infty} \gamma^t r_t$. A *policy* is a mapping $\pi : \mathcal{S} \rightsquigarrow \mathcal{A}$, and $\Pi$ is the set of all policies. Given a policy $\pi$, its *value function* $V^\pi : \mathcal{S} \to \mathbb{R}$ is the function where $V^\pi(s)$ is the expected future discounted reward when following $\pi$ from $s$, and its *Q-function* $Q^\pi : \mathcal{S} \times \mathcal{A} \to \mathbb{R} = \mathbb{E}_{S' \sim \tau(s,a)}[R(s,a,S') + \gamma \cdot V^\pi(S')]$. The *policy evaluation function* $J : \Pi \to \mathbb{R}$ is $J(\pi) = \mathbb{E}_{S_0 \sim \mu_0}[V^\pi(S_o)]$. If a policy maximises $J$, then we say that this policy is *optimal*. We denote optimal policies by $\pi^\star$, and their value function and $Q$-function by $V^\star$ and $Q^\star$. Moreover, given an MDP $\mathcal{M}$, we say that $\mathcal{M}$'s policy order is the ordering $\prec$ on $\Pi$ induced by $\pi_1 \prec \pi_2 \iff J(\pi_1) < J(\pi_2)$ for all $\pi_1, \pi_2$. For a more comprehensive overview, see Sutton & Barto (2018).

In this paper, we will say that a reward function $R$ is *trivial* if $J(\pi_1) = J(\pi_2)$ for all $\pi_1, \pi_2$. Moreover, we say that $R_1$ and $R_2$ are *equivalent* if $J_1(\pi_1) < J_1(\pi_2) \iff J_2(\pi_1) < J_2(\pi_2)$ for all $\pi_1, \pi_2$, and that they are *opposites* if $J_1(\pi_1) < J_1(\pi_2) \iff J_2(\pi_1) > J_2(\pi_2)$ for all $\pi_1, \pi_2$.

MORL problems are formalised using *Multi-Objective MDPs* (MOMDPs), which are tuples $\langle \mathcal{S}, \mathcal{A}, \tau, \mu_0, \vec{R}, \gamma \rangle$. The only place where MOMDPs differ from MDPs are $\vec{R}$, which is a function $\vec{R} : \mathcal{S} \times \mathcal{A} \times \mathcal{S} \rightsquigarrow \mathbb{R}^k$ that, for each transition $s, a, s'$, returns $k$ different rewards (for some $k$). We denote the reward function that returns the $i$'th component of $\vec{R}$ as $R_i$, and use $V_i^\pi$, $Q_i^\pi$, $J_i$, $G_i$, etc, to refer to its value functions, $Q$-functions, evaluation function, return function, etc. Since there may not be any single policy which maximises each component of $\vec{R}$, a MORL problem additionally needs a rule for how to combine and trade off each reward.

### 1.3 A Remark on "Tasks"

In order to determine if a given task can be expressed by Markovian reward functions, we must first determine what it means for a reward function to express a task. One answer to this question is to say that a task corresponds to a desired policy $\pi$, and that a reward function $R$ expresses the task if $\pi$ is optimal under $R$ (possibly with the additional requirement that $\pi$ is the *only* policy that is optimal under $R$). With this definition, we find that *any* task can be expressed as a Markovian reward function, at least as long as $\pi$ is stationary and deterministic (see Appendix B).

Another possible definition is to say that a task corresponds to an ordering $\prec$ on $\Pi$, which encodes a preference ordering over all policies, and that a reward function $R$ expresses the task if $J$ orders $\Pi$ according to $\prec$. It is primarily this latter definition that we will use in this paper. The main reason for this is that it often is impossible to find the optimal policy in complex environments. This means that it is not enough for $R$ to have the right optimal policy; it must also induce the right preferences between the (sub-optimal) policies that the policy optimisation algorithm actually considers. The only way to robustly ensure that this is the case is if $R$ induces the right policy ordering.

These are not the only two reasonable definitions. As mentioned previously, more definitions can be found in Abel et al. (2021).

## 2 Multi-Objective Reinforcement Learning

In this section, we examine the MORL setting. We first need a general definition of what a single-policy MORL problem is. Recall that a MOMDP $\langle \mathcal{S}, \mathcal{A}, \tau, \mu_0, \vec{R}, \gamma \rangle$ by itself has no one canonical objective to maximise. We therefore introduce the notion of a *MORL objective*:

**Definition 1.** *A **MORL objective** over $k$ rewards is a function $\mathcal{O}$ that takes $k$ policy evaluation functions $J_1 \ldots J_k$ and returns a (total) ordering $\prec_\mathcal{O}$ over the set of all policies $\Pi$.*

Given a MOMDP $\mathcal{M} = \langle \mathcal{S}, \mathcal{A}, \tau, \mu_0, \vec{R}, \gamma \rangle$, a MORL objective $\mathcal{O}$ gives us an ordering over $\Pi$ that tells us when a policy is preferred over another. We use $\prec_\mathcal{O}^\mathcal{M}$ to denote the policy ordering that is obtained when we apply $\mathcal{O}$ to $\mathcal{M}$'s policy evaluation functions. For the purposes of this paper, we will not need to impose any further requirements on $\prec_\mathcal{O}$. For example, we will not insist that $\prec_\mathcal{O}$ must have a greatest element in $\Pi$, or that $\pi_1 \prec_O \pi_2$ whenever $\pi_2$ is a Pareto improvement over $\pi_1$, etc, even though a reasonable MORL objective presumably would have these properties. We next give a few examples of some interesting MORL objectives:

**Definition 2.** *Given $J_1 \ldots J_k$, the **LexMax** objective $\prec_{Lex}$ is given by $\pi_1 \prec_{Lex} \pi_2$ if and only if there is an $i \in \{1 \ldots m\}$ such that $J_i(\pi_1) < J_i(\pi_2)$, and $J_j(\pi_1) = J_j(\pi_2)$ for $j < i$.*

**Definition 3.** *Given* $J_1 \ldots J_k$, *the **MaxMin** objective* $\prec_{Min}$ *is given by* $\pi_1 \prec_{Min} \pi_2 \iff$ $\min_i J_i(\pi_1) < \min_i J_i(\pi_2)$.

**Definition 4.** *Given* $J_1 \ldots J_k$ *and some* $c_1 \ldots c_m \in \mathbb{R}$, *the **MaxSat** objective* $\prec_{Sat}$ *is given by* $\pi_1 \prec_{Sat} \pi_2$ *if and only if the number of rewards that satisfy* $J_i(\pi_1) \geq c_i$ *is larger than the number of rewards that satisfy* $J_i(\pi_2) \geq c_i$.

**Definition 5.** *Given* $J_1, J_2$ *and some* $c \in \mathbb{R}$, *the **ConSat** objective* $\prec_{Con}$ *is given by* $\pi_1 \prec_{Con} \pi_2$ *if and only if either* $J_1(\pi_1) < c$ *and* $J_1(\pi_1) < J_1(\pi_2)$, *or if* $J_1(\pi_1), J_1(\pi_2) \geq c$ *and* $J_2(\pi_1) < J_2(\pi_2)$.

In other words, the LexMax objective has *lexicographic* preferences over $R_1 \ldots R_m$, so that policies are first ordered by their expected discounted $R_1$-reward, and then policies that obtain the same expected discounted $R_1$-reward are ordered by their expected discounted $R_2$-reward, and so on. The MaxMin objective orders policies by their *worst* performance according to any of $R_1 \ldots R_m$ (which could be used to obtain worst-case guarantees). The MaxSat objective only cares about whether a policy reaches a certain *threshold* for each reward, and ranks policies based on how many thresholds they reach. The ConSat objective wants to maximise $J_2$, but under the constraint that $J_1$ reaches a certain threshold. These MORL objectives are simply a short list of illustrative examples, demonstrating the flexibility of the framework. A few more examples are given in Appendix D. We next need to define what it means to *reduce* a MORL problem to a (scalar) RL problem:

**Definition 6.** *A MOMDP* $\mathcal{M} = \langle \mathcal{S}, \mathcal{A}, \tau, \mu_0, \vec{R}, \gamma \rangle$ *with objective* $\mathcal{O}$ *is **equivalent** to the MDP* $\tilde{\mathcal{M}} = \langle \mathcal{S}, \mathcal{A}, \tau, \mu_0, \tilde{\mathbb{R}}, \gamma \rangle$ *if and only if* $\tilde{M}$'s *policy order is* $\prec_{\mathcal{O}}^{\mathcal{M}}$.

Note that $\tilde{\mathcal{M}}$ must have the same states, actions, transition function, initial state distribution, and discount factor, as $\mathcal{M}$. This definition therefore says that $\mathcal{M}$ with $\mathcal{O}$ is equivalent to $\tilde{\mathcal{M}}$ if $\tilde{M}$ is given by replacing $\vec{R} = \langle R_1 \ldots R_k \rangle$ with a single reward function $\tilde{R}$, and $\tilde{R}$ induces the same preferences between all policies as $\mathcal{O}(J_1 \ldots J_k)$. We can now derive necessary and sufficient conditions for when a MORL problem can be reduced to a scalar-reward RL problem.

**Theorem 1.** *If a MOMDP* $\mathcal{M} = \langle \mathcal{S}, \mathcal{A}, \tau, \mu_0, \vec{R}, \gamma \rangle$ *with objective* $\mathcal{O}$ *is equivalent to an MDP* $\tilde{M} = \langle \mathcal{S}, \mathcal{A}, \tau, \mu_0, \tilde{\mathbb{R}}, \gamma \rangle$, *then* $\tilde{J}(\pi) = \sum_{i=1}^{k} w_i \cdot J_i(\pi)$ *for some* $w_1 \ldots w_k \in \mathbb{R}$. *Moreover,* $\mathcal{M}$ *with* $\mathcal{O}$ *is also equivalent to the MDP with reward* $R(s, a, s') = \sum_{i=1}^{k} w_i \cdot R_i(s, a, s')$.

*Proof.* Suppose $\mathcal{M}$ with $\mathcal{O}$ is equivalent to an MDP $\tilde{\mathcal{M}} = \langle \mathcal{S}, \mathcal{A}, \tau, \mu_0, \tilde{\mathbb{R}}, \gamma \rangle$. First, let $m : \Pi \to \mathbb{R}^{|\mathcal{S}||\mathcal{A}|}$ be the function that maps each policy $\pi$ to the $|\mathcal{S}||\mathcal{A}|$-dimensional vector where

$$m(\pi)[s, a] = \sum_{t=0}^{\infty} \gamma^t \mathbb{P}_{\xi \sim \pi}(S_t = s, A_t = a).$$

Moreover, for a reward function $R$, let $\vec{R} \in \mathbb{R}^{|\mathcal{S}||\mathcal{A}|}$ be the $|\mathcal{S}||\mathcal{A}|$-dimensional vector where

$$\vec{R}[s, a] = \mathbb{E}_{S' \sim \tau(s,a)}[R(s, a, S')].$$

Note that we now have that $J(\pi) = m(\pi) \cdot \vec{R}$, for any reward function $R$. Recall also that multiplication by an $|\mathcal{S}||\mathcal{A}|$-dimensional vector induces a linear function over $\mathbb{R}^{|\mathcal{S}||\mathcal{A}|}$. This means that, for any reward function $R$, we can express its policy evaluation function $J : \Pi \to \mathbb{R}$ as $L \circ m$, where $L$ is a linear function. In particular, $\tilde{J} = \tilde{L} \circ m$, and $J_i = L_i \circ m$ for each of $R_i \in \vec{R}$.

From the definition of MORL objectives, we have that $\tilde{J}(\pi)$ is a function of $J_1(\pi) \ldots J_k(\pi)$. This, in turn, means that $\tilde{L}(v)$ is a function of $L_1(v) \ldots L_k(v)$, for any $v \in \text{Im}(m)$. Let $M$ be the $(|S||A| \times k)$-dimensional matrix that maps each vector $v \in \mathbb{R}^{|\mathcal{S}||\mathcal{A}|}$ to $\langle L_1(v), \ldots, L_k(v) \rangle$ (in other words, the matrix whose rows are $\vec{R}_1 \ldots \vec{R}_k$). Since $\tilde{L}(v)$ is a function of $L_1(v) \ldots L_k(v)$, we have that $\tilde{L}$ can be expressed as $f \circ M$ for some function $f$. Since $\tilde{L}$ is a linear function, and since $M$ is a linear transformation, we that $f$ must be a linear function as well. This means that there are $w_1 \ldots w_k \in \mathbb{R}^k$ such that $f(x) = \sum_{i=1}^{k} w_i \cdot x_i$, which implies that $\tilde{L}(v) = \sum_{i=1}^{m} w_i \cdot L_i(v)$, and further that $\tilde{J}(\pi) = \sum_{i=1}^{k} w_i \cdot J_i(\pi)$. This completes the first part.

Next, let $R(s, a, s') = \sum_{i_1}^{k} w_i \cdot R_i(s, a, s')$. Straightforward algebra shows that $J(\pi) = \sum_{i=1}^{k} w_i \cdot J_i(\pi)$. Now, since $J = \tilde{J}$, and since $\mathcal{M}$ with $\mathcal{O}$ is equivalent to $\tilde{\mathcal{M}}$, we have that $\mathcal{M}$ with $\mathcal{O}$ is equivalent to the MDP with reward $R$. This completes the second part. $\qquad\square$

This theorem effectively tells us that *only linear* MORL objectives can be represented using scalar-reward RL! This imposes a harsh limitation on what kinds of tasks can be encoded using scalar rewards. Theorem 1 also has the following corollary, which is useful for demonstrating when some MORL objective cannot be expressed using scalar reward functions. Given an ordering $\prec$ over $\Pi$ dependent on some evaluation functions $J_1 \ldots J_k$, we say that a function $U : \Pi \to \mathbb{R}$ *represents* $\prec$ if $U(\pi_1) < U(\pi_2) \iff \pi_1 \prec \pi_2$. We say that $U$ is a *linear representation* if $U(\pi) = f(\sum_{i=1}^{k} w_i \cdot J_i(\pi))$ for some $w_1 \ldots w_k \in \mathbb{R}$ and some $f$ that is strictly monotonic.

**Corollary 1.** *If $\mathcal{O}(J_1 \ldots J_k)$ has a non-linear representation $U$, and $\mathcal{M}$ is a MOMDP whose $J$-functions are $J_1 \ldots J_k$, then $\mathcal{M}$ with $\mathcal{O}$ is not equivalent to any MDP.*

*Proof.* Assume for contradiction that $\mathcal{M}$ with $\mathcal{O}$ is equivalent the MDP $\tilde{\mathcal{M}} = \langle \mathcal{S}, \mathcal{A}, \tau, \mu_0, \tilde{\mathbb{R}}, \gamma \rangle$. Then $\tilde{J}$ represents $\mathcal{O}(J_1 \ldots J_k)$, and this in turn means that $U$ must be strictly monotonic in $\tilde{J}$. Moreover, Theorem 1 implies that $\tilde{J} = \sum_{i=0}^{k} w_i \cdot J_i$ for some $w_1 \ldots w_k \in \mathbb{R}^k$. However, this contradicts our assumptions. $\qquad\square$

Therefore, we can prove that $\mathcal{M}$ with $\mathcal{O}$ is not equivalent to any MDP by finding a non-linear representation of $\prec_{\mathcal{O}}^{\mathcal{M}}$. We will now show that none of the MORL objectives given in Definition 2-5 can be expressed using single-objective RL, except in a few degenerate edge cases.

**Theorem 2.** *There is no MDP equivalent to $\mathcal{M}$ with LexMax, as long as $\mathcal{M}$ has at least two reward functions that are neither trivial, equivalent, or opposites.*

*Proof.* Suppose $\mathcal{M}$ with LexMax is equivalent to $\tilde{\mathcal{M}} = \langle \mathcal{S}, \mathcal{A}, \tau, \mu_0, \tilde{\mathbb{R}}, \gamma \rangle$. Let $i$ be the smallest number such that $R_i$ is non-trivial, and let $j$ be the smallest number greater than $i$ such that $R_j$ is non-trivial, and not equivalent to or opposite of $R_i$. Then there are $\pi_1, \pi_2$ such that $J_i(\pi_1) = J_i(\pi_2)$ and $J_j(\pi_1) < J_j(\pi_2)$, which means that $\pi_1 \prec_{\text{Lex}}^{\mathcal{M}} \pi_2$. Moreover, since $\tilde{J}$ represents $\prec_{\text{Lex}}^{\mathcal{M}}$, it follows that there are no $\pi, \pi'$ such that $J_i(\pi) < J_i(\pi')$ and $\tilde{J}(\pi) > \tilde{J}(\pi')$. Then Theorem 1 in Skalse et al. (2022c) implies that $R_i$ is equivalent to $\tilde{R}$. However, then $\tilde{J}(\pi_1) = \tilde{J}(\pi_2)$, which means that $\tilde{J}$ cannot represent $\prec_{\text{Lex}}^{\mathcal{M}}$. $\qquad\square$

**Theorem 3.** *There is no MDP equivalent to $\mathcal{M}$ with MaxMin, unless $\mathcal{M}$ has a reward function $R_i$ such that $J_i(\pi) \leq J_j(\pi)$ for all $j \in \{1 \ldots k\}$ and all $\pi$.*

*Proof.* $\mathcal{O}_{\text{Min}}^{\mathcal{M}}$ is represented by the function $U(\pi) = \min_i J_i(\pi)$. Moreover, if $\mathcal{M}$ has no reward function $R_i$ such that $J_i(\pi) \leq J_j(\pi)$ for all $j \in \{1 \ldots k\}$ and all $\pi$ then this representation is non-linear. Corollary 1 then implies that $\mathcal{M}$ with MaxMin is not equivalent to any MDP. $\qquad\square$

**Theorem 4.** *There is no MDP equivalent to $\mathcal{M}$ with MaxSat, as long as $\mathcal{M}$ has at least one reward $R_i$ where $J_i(\pi_1) < c_i$ and $J_i(\pi_2) \geq c_i$ for some $\pi_1, \pi_2 \in \Pi$.*

*Proof.* Note that $\text{MaxSat}(\mathcal{M})$ is represented by the function $U(\pi) = \sum_{i=1}^{k} \mathbb{1}[J_i(\pi) \geq c_i]$, where $\mathbb{1}[J_i(\pi) \geq c_i]$ is the function that is equal to 1 when $J_i(\pi) \geq c_i$, and 0 otherwise. Moreover, $U$ is not strictly monotonic in any function that is linear in $J_1 \ldots J_k$. Corollary 1 thus implies that $\mathcal{M}$ with MaxSat is not equivalent to any MDP. $\qquad\square$

**Theorem 5.** *There is no MDP equivalent to $\mathcal{M}$ with ConSat, unless either $R_1$ and $R_2$ are equivalent, or $\max_\pi J_1(\pi) \leq c$.*

*Proof.* $\mathcal{O}_{\text{Con}}^{\mathcal{M}}$ is represented by $U(\pi) = \{J_1(\pi)$ if $J_1(\pi) \leq c$, else $J_2(\pi) - \min_\pi J_2(\pi) + c\}$. Moreover, this representation is non-linear, unless either $R_1$ and $R_2$ are equivalent, or $\max_\pi J_1(\pi) \leq c$. Corollary 1 then implies that $\mathcal{M}$ with ConSat is not equivalent to any MDP. $\qquad\square$

Theorem 2-5 show that none of the MORL objectives given in Definition 2-5 can be expressed using single-objective RL, except in a few degenerate cases where those MORL objectives are uninteresting. This demonstrates that there is no satisfactory way to reduce MORL problems to scalar-reward RL (and hence that scalar RL is unable to express many natural task specifications).

## 3 RISK-SENSITIVE REINFORCEMENT LEARNING

The next area we will look at is that of *risk-sensitive* reinforcement learning. An ordinary RL agent tries to maximise the *expectation* of its reward function. However, there are many cases where it is natural to want the agent to be *risk-averse*. In economics, risk-aversion is typically modelled by using utility functions $U(c)$ that are concave in some relevant quantity $c$ (which might be money, for example). A natural question is then whether a similar trick may be used with reward functions? That is, given a reward function $R_1$ and a concave function $f$, can we construct a reward function $R_2$ such that $G_2(\xi) = f(G_1(\xi))$ for all trajectories $\xi$? We will examine this question.

Some of the most common risk-averse utility functions includes *exponential utility*, *isoelastic utility*, and *quadratic utility*. The exponential utility function is given by $U(c) = -e^{\alpha c}$, where $\alpha > 0$ is a parameter controlling the degree of risk aversion. The isoelastic utility function is given by $U(c) = c^{1-\alpha}$, for $\alpha > 0, \alpha \neq 1$, or by $U(c) = \ln(c)$ (corresponding to the case when $\alpha = 1$). The quadratic utility function is given by $U(c) = c - \alpha c^2$, where $\alpha > 0$. Since this function is decreasing for sufficiently large $c$, its domain is typically restricted to $(-\infty, 1/2\alpha]$. We will examine each of these, and show that none of them can be expressed using reward functions.

In this section, we will consider the domain of $G$ to be the set of all coherent trajectories, *not* the set of trajectories which are possible under some transition function $\tau$. In other words, we consider the set of all trajectories to be $(\mathcal{S} \times \mathcal{A})^\omega$. The reason for this is that we do not want to presume any prior knowledge of the environment. If we restrict the set of trajectories we consider, then some risk-averse utility functions can become possible to express (consider the case of a tree-shaped MDP, for example). Finally, we will say that $R$ is *constant* if it has a constant value for all $s, a, s'$.

To prove our results, we will make use of three lemmas. The proofs of these lemmas are fairly long, but not very illuminating, and so we have relegated them to Appendix A.

**Lemma 1.** *If $R$ is non-constant, then for any state $s$ there exists trajectories $\zeta_1, \zeta_2, \zeta_3$ starting in $s$ such that $G(\zeta_1) \neq G(\zeta_2)$, $G(\zeta_2) \neq G(\zeta_3)$, and $G(\zeta_1) \neq G(\zeta_3)$.*

**Lemma 2.** *If $G_2(\xi) = f(G_1(\xi))$ for all $\xi$ and some $f$, then for any transition $\langle s, a, s' \rangle$ and any trajectory $\zeta$ starting in $s'$, $R_2(s, a, s') = f(R_1(s, a, s') + \gamma G_1(\zeta)) - \gamma f(G_1(\zeta))$.*

**Lemma 3.** *For any non-constant reward $R_1$ and any $f$ that is injective on $\mathrm{range}(G_1)$, if for any $y \in \mathrm{range}(R_1)$ and any $\gamma \in (0, 1)$ there are at most two distinct $x_1, x_2$ such that $f(y + \gamma x_1) - \gamma f(x_1) = f(y + \gamma x_2) - \gamma f(x_2)$ then there is no reward $R_2$ such that $G_2(\xi) = f(G_1(\xi))$ for all $\xi$.*

Using these lemmas, we can now derive our main results:

**Theorem 6.** *For any non-constant reward function $R_1$ and any constant $\alpha \neq 0$, there is no reward function $R_2$ such that $G_2(\xi) = -e^{\alpha G_1(\xi)}$ for all valid trajectories $\xi$.*

*Proof.* With $f(x) = -e^{\alpha x}$, the expression in Lemma 3 becomes $-e^{\alpha(y+\gamma x)} + \gamma e^{\alpha x}$. The derivative of this expression with respect to $x$ is $\gamma\alpha(-e^{\alpha(y+\gamma x)} + e^{\alpha x})$, which has only one root when $\gamma \neq 0$ and $\alpha \neq 0$. This means that there can be at most two distinct values $x_1, x_2$ such that $-e^{\alpha(y+\gamma x_1)} + \gamma e^{\alpha x_1} = -e^{\alpha(y+\gamma x_2)} + \gamma e^{\alpha x_2}$. Since $-e^{\alpha x}$ is injective, we can thus apply Lemma 3, which completes the proof. $\square$

**Theorem 7.** *For any non-constant reward function $R_1$ and any constant $\alpha > 0$, $\alpha \neq 1$, there is no reward function $R_2$ such that $G_2(\xi) = G_1(\xi)^{1-\alpha}$ for all valid trajectories $\xi$.*

*Proof.* With $f(x) = x^{1-\alpha}$, the expression in Lemma 3 becomes $(y + \gamma x)^{(1-\alpha)} - \gamma x^{1-\alpha}$. The derivative of this expression with respect to $x$ is $\gamma(\alpha - 1)(x^{-\alpha} - (\gamma x + y)^{-\alpha})$, which has only one root when $\gamma \neq 0$ and $\alpha \notin \{0, 1\}$. This means that there can be at most two distinct values $x_1, x_2$ such that $(y + \gamma x_1)^{(1-\alpha)} - \gamma x_1^{1-\alpha} = (y + \gamma x_2)^{(1-\alpha)} - \gamma x_2^{1-\alpha}$. Since $x^{1-\alpha}$ is injective, we can thus apply Lemma 3, which completes the proof. $\square$

**Theorem 8.** *For any non-constant reward function $R_1$, there is no reward function $R_2$ such that $G_2(\xi) = \ln(G_1(\xi))$ for all valid trajectories $\xi$.*

*Proof.* With $f(x) = \ln(x)$, the expression in Lemma 3 becomes $\ln(y+\gamma x) - \gamma \ln(x)$. The derivative of this expression with respect to $x$ is $\gamma(1/(y + \gamma x) - 1/x)$, which has only one root when $\gamma \neq 0$. Since $\ln(x)$ is injective, we can thus apply Lemma 3, which completes the proof. $\qquad\square$

**Theorem 9.** *For any non-constant reward function $R_1$ and any $\alpha > 0$ where $\max_\xi G_1(\xi) \leq \frac{1}{2\alpha}$, there is no reward function $R_2$ such that $G_2(\xi) = G_1(\xi) - \alpha G_1(\xi)^2$ for all $\xi$.*

*Proof.* With $f(x) = x - \alpha x^2$, the expression in Lemma 3 becomes $y + \gamma x - \alpha(y + \gamma x)^2$. This is a second-degree polynomial, which means that there can be at most two distinct values $x_1, x_2$ such that $y + \gamma x_1 - \alpha(y + \gamma x_1)^2 = y + \gamma x_2 - \alpha(y + \gamma x_2)^2$. Moreover, if $\max_\xi G_1(\xi) \leq \frac{1}{2\alpha}$ then $f(x) = x - \alpha x^2$ is injective on $\mathrm{range}(G_1)$. We can thus apply Lemma 3. $\qquad\square$

We can thus see that Lemma 3 is quite flexible. It allows us to rule out many modifications to $G$ as impossible, including all the standard risk-averse utility functions. It would be desirable to strengthen these results, and provide necessary and sufficient conditions for when it is possible to construct a reward $R_2$ such that $G_2(\xi) = f(G_1(\xi))$ for some function $f$ and some (non-constant) reward $R_1$. We consider this to be an important question for further work.

## 4 MODAL REINFORCEMENT LEARNING

The final class of tasks we will examine is one which we have decided to refer to as *modal* tasks. Before we give a formal definition of this class, we will first provide some intuition. In analytic philosophy, a distinction is made between *categorical* facts and *modal* facts. In short, categorical facts only concern what is true in actuality, whereas modal facts concern what must be true, could have been true, or cannot be true, etc. For example, it is a categorical fact that the Eiffel Tower is brown, and a modal fact that it *could have had* a different colour. It is (arguably) a categorical fact that the number 3 is prime, and a modal fact that it *could not have been* otherwise. To give another example, there is a difference between stating that *nothing can travel faster than light* and that *nothing does travel faster than light* – the former statement, which is modal, is stronger than the latter, which is categorical. One can further distinguish between different kinds of possibility (e.g. logical vs physical possibility, etc), and discussions about modality also involves topics such as *causality* and *counterfactuals*, etc. A complete treatment of this subject is far beyond the scope of this paper, but for an overview, see e.g. Menzel (2021).

Modality does of course relate to *modal logic*, but it also relates to *temporal logic*. In particular, computational tree logic (CTL), and its extensions, can express many modal statements.

The intuition behind this section is that a reward function always is expressed in terms of categorical facts, whereas many tasks are naturally expressed in terms of modal facts. For example, consider an instruction such as "you should always be *able* to return to the start state". This instruction seems quite reasonable, but it is not obvious how to translate it into a reward function. Note that this instruction is not telling the agent to *actually* return to the start state, it merely says that it should maintain the *ability* to do so. To give a few other examples, consider instructions such as "you should never enter a state from which it is *possible* to quickly enter an unsafe state", "you should always be *able* to press the emergency shutdown button", or "you should never enter a state where you would be *unable* to receive a feedback signal". These instructions all seem very reasonable, and they are expressed in terms of what should be *possible* or *impossible* along the trajectory of the agent, rather than in terms of what in fact occurs along that trajectory. Given this background motivation, we can now give a formal definition of modal tasks:

**Definition 7.** *Given a set of states $\mathcal{S}$ and a set of actions $\mathcal{A}$, a **modal reward function** $R^\diamond$ is a function $R^\diamond : \mathcal{S} \times \mathcal{A} \times \mathcal{S} \times (\mathcal{S} \times \mathcal{A} \rightsquigarrow \mathcal{S}) \rightarrow \mathbb{R}$ which takes two states $s, s' \in \mathcal{S}$, an action $a \in \mathcal{A}$, and a transition function $\tau$ over $\mathcal{S}$ and $\mathcal{A}$, and returns a real number.*

$R^\diamond(s, a, s', \tau)$ is the reward that is obtained when transitioning from state $s$ to $s'$ using action $a$ in an environment whose transition function is $\tau$. Here we allow $R^\diamond$ an unrestricted dependence on $\tau$, to make our results as general as possible, even if a practical algorithm for solving modal tasks presumably would require restrictions on what this dependence can look like (see Appendix E). Modal reward functions can be used to express instructions such as those we gave above. For example, a simple case might be "you get 1 reward if you reach this goal state, and -1 reward if

you ever enter a state from which you cannot reach the initial state". This reward depends on the transition function, because the transition function determines from which states you can reach the initial state. As usual, $R^\diamond$ then induces a $Q$-function $Q^\diamond$, value function $V^\diamond$, and evaluation function $J^\diamond$, etc. We say that a modal reward $R^\diamond$ and an ordinary reward $R$ are *contingently equivalent* given a transition function $\tau$ if $J^\diamond$ and $J$ induce the same ordering of policies given $\tau$, and that they are *robustly equivalent* if $J^\diamond$ and $J$ induce the same ordering of policies for all $\tau$. We use $R^\diamond_\tau$ to denote the reward function $R^\diamond_\tau(s, a, s') = R^\diamond(s, a, s', \tau)$. We will also use the following definition:

**Definition 8.** *A modal reward function $R^\diamond$ is **trivial** if there is a reward function $R$ such that for all $\tau$, $R$ and $R^\diamond_\tau$ have the same policy ordering under $\tau$.*

The intuition here is that a trivial modal reward function does not actually depends on $\tau$ in any important sense. Note that this is *not* necessarily to say that $R^\diamond_\tau = R$ for all $\tau$. For example, it could be the case that $R^\diamond_\tau$ is a *scaled* version of $R$, or that $R^\diamond_\tau$ and $R$ differ by *potential shaping* Ng et al. (1999), or that $R^\diamond_\tau$ is modified in a way such that $\mathbb{E}_{S' \sim \tau(s,a)}[R^\diamond_\tau(s, a, S')] = \mathbb{E}_{S' \sim \tau(s,a)}[R(s, a, S')]$, since none of these differences affect the policy ordering.

**Theorem 10.** *For any modal reward $R^\diamond$ and any transition function $\tau$, there exists a reward function $R$ that is contingently equivalent to $R^\diamond$ given $\tau$. Moreover, unless $R^\diamond$ is trivial, there is no reward function that is robustly equivalent to $R^\diamond$.*

*Proof.* This is straightforward. For the first part, simply let $R(s, a, s') = R^\diamond(s, a, s', \tau)$. The second part is immediate from the definition of trivial modal reward functions. $\square$

In other words, every modal task can be expressed with ordinary reward function in each particular environment, but no reward function expresses a (non-trivial) modal task in all environments. Is this enough? We argue that it is not, because the construction of $R^\diamond_\tau$ will invariably be laborious, and require detailed knowledge of the environment. For example, consider the task "you should always be able to return to the start state"; here, constructing $R^\diamond_\tau$ would amount to manually enumerating all the states from which the start state is reachable. This is very much against the spirit of reinforcement learning, where much of the point is that we want to be able to specify tasks which can be pursued in *unknown* environments. In short, a method which requires a model of the environment is arguably not a reinforcement learning method. We thus argue that reward functions are unable to capture modal tasks in a satisfactory way.

One remaining question might be why one would want to express instructions for reinforcement learning agents in terms of modal properties. After all, what benefit is there to the instruction "never enter a state from which it is possible to quickly enter an unsafe state" over the instruction "never enter an unsafe state"? One reason is that the former task might lead to behaviour that is more robust to changes in the environment. For example, if an RL agent is trained in a simulated environment, and deployed in the real world, then it seems like it would be preferable to tell the agent to avoid *risky* states, rather than *unsafe* states, since imperfections in the simulation could lead to an underestimation of the risk involved. Another example is the existing work on avoiding side effects (Krakovna et al., 2018; 2020; Turner et al., 2020), which it is natural to express in modal terms. This work can be viewed as being aimed at making the behaviour of an RL agent more robust to misspecification of the reward function.

## 5 Solving "Inexpressible" Tasks

We have pointed to three classes of tasks which cannot be expressed using reward functions (namely multi-objective tasks, risk-sensitive tasks, and modal tasks). A natural next question is whether these tasks *could* be solved using RL, or whether only the tasks which correspond to Markovian reward functions can be effectively learnt? We discuss this issue below.

In short, it is possible to design RL algorithms for tasks in each of these categories. Multi-objective reinforcement learning is well-explored, with many existing algorithms (see Section 1.1). Most of these algorithms are designed to solve a specific MORL objective; for example, Skalse et al. (2022b) solve the LexMax objective, and Tessler et al. (2019) solve the ConSat objective. There is (to the best of our knowledge) not yet any algorithm for the solving e.g. the MaxMin objective, but there is no good reason to believe that such an algorithm could not be made. Similarly, there are existing

algorithms for risk-sensitive RL (e.g. Chow et al. (2017)), and even algorithms that solve certain modal tasks (Krakovna et al., 2018; 2020; Turner et al., 2020; Wang et al., 2020).

It should also be possible to design algorithms which can flexibly solve many different tasks from the classes we have discussed (instead of having to be designed for just one particular task). For example, suppose a MORL objective can be represented by a function $U : \mathbb{R}^k \to \mathbb{R}$, such that $\pi_1 \prec \pi_2$ when $U(J_1(\pi_1) \dots J_k(\pi_1)) < U(J_1(\pi_2) \dots J_k(\pi_2))$, and that $U$ is *differentiable*. We give a few examples of such objectives in Appendix D, including e.g. a "soft" version of MaxMin. With such an objective, if we have a policy $\pi$ that is differentiable with respect to some parameters $\theta$, then it should be possible to compute the gradient of $U(J_1(\pi) \dots J_k(\pi))$ with respect to $\theta$, and then use a policy gradient method to increase $U$. This means that it should be possible to design an actor-critic algorithm which can solve any differentiable MORL objective. We consider the development and evaluation of such methods to be a promising direction for further work.

We outline a possible approach for solving a wide class of modal tasks in Appendix E.

## 6 DISCUSSION

In this paper, we have studied the ability of Markovian reward functions to express different kinds of problems. We have looked at three classes of tasks; multi-objective tasks, risk-sensitive tasks, and modal tasks, and found that Markovian reward functions are unable to express most of the tasks in each of these three classes. We have also provided necessary and sufficient conditions for when a single-policy MORL problem can be expressed using a single reward function (which, as it turns out, is almost never), and also drawn attention to a class of tasks which have just barely been explored previously (namely modal tasks). Finally, we have also shown that many of these problems still can be solved with RL, and even outlined some methods for how to extend these solutions.

There are several ways to extend our work. First of all, while we have given many examples of tasks which cannot be formalised using Markovian reward functions, we have not given a general characterisation of what reward functions are or are not able to express. It would be very desirable to have a set of intuitive necessary and sufficient conditions, which exactly describe those policy orderings that can be expressed using reward functions, similar to what the VNM axioms provide in the case of utility functions. We outline some initial steps towards such a characterisation in Appendix B. Note that the VNM axioms themselves cannot be directly applied to RL, see Appendix C. Additionally, it would also be desirable to provide necessary and sufficient conditions for when it is possible to construct a reward $R_2$ such that $G_2(\xi) = f(G_1(\xi))$ for some function $f$ and some (non-constant) reward $R_1$, as we discussed at the end of Section 3.

Our work also provides a strong motivation for developing more RL algorithms that can learn tasks which cannot be expressed using Markovian reward functions. There are several ways to to this. In section 5, we outline an approach for learning any differentiable MORL objective using policy gradients, and in Appendix E, we outline an approach for learning a large class of modal tasks. It would also be very interesting to explore more general ways to express RL tasks, and study their expressivity. For example, it would be interesting to know if (and to what extent) MORL tasks can be expressed using reward machines (Icarte et al., 2022), and similar.

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

# A  PROOFS OF LEMMAS

In this Appendix, we provide the proofs of the lemmas from Section 3.

**Lemma 1.** *If $R$ is non-constant, then for any state $s$ there exists trajectories $\zeta_1, \zeta_2, \zeta_3$ starting in $s$ such that $G(\zeta_1) \neq G(\zeta_2)$, $G(\zeta_2) \neq G(\zeta_3)$, and $G(\zeta_1) \neq G(\zeta_3)$.*

*Proof.* First note that if $R$ is non-constant, then there must be *some* state $s$ and some trajectories $\xi_1, \xi_2$ starting in $s$ such that $G(\xi_1) \neq G(\xi_2)$ (this follows from Theorem 3.8 in Skalse et al. (2022a)). We will establish that there is a $\xi_3$ starting in $s$ such that $G(\xi_3) \neq G(\xi_1)$ and $G(\xi_3) \neq G(\xi_2)$, and then show that this implies that such trajectories exist for *all* states.

Suppose for contradiction that for any $\xi_3$ starting in $s$, either $G(\xi_3) = G(\xi_1)$ or $G(\xi_3) = G(\xi_2)$. Consider a transition $\langle s, a, s \rangle$, and let $\zeta_1 = \langle s, a, s \rangle + \xi_1$ and $\zeta_2 = \langle s, a, s \rangle + \xi_2$; we will do a case enumeration, and show that either $G(\zeta_1)$ or $G(\zeta_2)$ must be distinct from both $G(\xi_1)$ and $G(\xi_2)$. Note that $G(\zeta_1) = R(s, a, s) + \gamma G(\xi_1)$ and $G(\zeta_2) = R(s, a, s) + \gamma G(\xi_2)$.

**Case 1**: $G(\zeta_1) = G(\xi_1)$, $G(\zeta_2) = G(\xi_2)$. If $R(s, a, s) + \gamma G(\xi_1) = G(\xi_1)$ then $R(s, a, s) = (1 - \gamma)G(\xi_1)$, and similarly, if $R(s, a, s) + \gamma G(\xi_2) = G(\xi_2)$ then $R(s, a, s) = (1 - \gamma)G(\xi_2)$. This is a contradiction, since $G(\xi_1) \neq G(\xi_2)$ and $\gamma \neq 1$.

**Case 2**: $G(\zeta_1) = G(\zeta_2) = G(\xi_1)$. If $R(s, a, s) + \gamma G(\xi_1) = G(\xi_1)$ then $R(s, a, s) = (1 - \gamma)G(\xi_1)$. Using $R(s, a, s) + \gamma G(\xi_2) = G(\xi_1)$, we get $(1 - \gamma)G(\xi_1) + \gamma G(\xi_2) = \gamma G(\xi_1)$. By rearranging, we get $\gamma(G(\xi_1) - G(\xi_2)) = 0$. This is a contradiction, since $G(\xi_1) \neq G(\xi_2)$ and $\gamma \neq 0$.

**Case 3**: $G(\zeta_1) = G(\zeta_2) = G(\xi_2)$. This is analogous to Case 2.

**Case 4**: $G(\zeta_1) = G(\xi_2)$, $G(\zeta_2) = G(\xi_1)$. If $R(s, a, s) + \gamma G(\xi_1) = G(\xi_2)$ then $R(s, a, s) = G(\xi_2) - \gamma G(\xi_2)$, and similarly, if $R(s, a, s) + \gamma G(\xi_2) = G(\xi_1)$ then $R(s, a, s) = G(\xi_1) - \gamma G(\xi_2)$. Combining this, and rearranging, gives $(1 + \gamma)G(\xi_1) = (1 + \gamma)G(\xi_2)$. This is a contradiction, since $G(\xi_1) \neq G(\xi_2)$ and $\gamma \neq -1$.

This exhausts all cases, which means that if $R$ is non-constant, then there must be some state $s$ and some trajectories $\zeta_1, \zeta_2, \zeta_3$ starting in $s$ such that $G(\zeta_1) \neq G(\zeta_2)$, $G(\zeta_2) \neq G(\zeta_3)$, and $G(\zeta_1) \neq G(\zeta_3)$. Finally, note that this means that we can construct such trajectories for *any* state $s'$, by simply composing a transition $\langle s', a, s \rangle$ with each of $\zeta_1, \zeta_2, \zeta_3$. □

**Lemma 2.** *If $G_2(\xi) = f(G_1(\xi))$ for all $\xi$ and some $f$, then for any transition $\langle s, a, s' \rangle$ and any trajectory $\zeta$ starting in $s'$, $R_2(s, a, s') = f(R_1(s, a, s') + \gamma G_1(\zeta)) - \gamma f(G_1(\zeta))$.*

*Proof.* Suppose that $G_2(\xi) = f(G_1(\xi))$ for all trajectories $\xi$. Let $\langle s, a, s' \rangle$ be an arbitrary transition, let $\zeta$ be an arbitrary trajectory starting in $s'$, and let $\xi = \langle s, a, s' \rangle + \zeta$. We have that $G_2(\xi) = R_2(s, a, s') + \gamma G_2(\zeta)$, and also that $G_2(\xi) = f(G_1(\xi))$, which implies that

$$R_2(s, a, s') + \gamma G_2(\zeta) = f(G_1(\xi)).$$

Since $G_1(\xi) = R_1(s, a, s') + \gamma G_1(\zeta)$, this implies that

$$R_2(s, a, s') + \gamma G_2(\zeta) = f(R_1(s, a, s') + \gamma G_1(\zeta)).$$

By using the fact that $G_2(\zeta) = f(G_1(\zeta))$, and rearranging, we get that

$$R_2(s, a, s') = f(R_1(s, a, s') + \gamma G_1(\zeta)) - \gamma f(G_1(\zeta)).$$

Since $\langle s, a, s' \rangle$ and $\zeta$ were chosen arbitrarily, this completes the proof. □

**Lemma 3.** *For any non-constant reward $R_1$ and any $f$ that is injective on $\mathrm{range}(G_1)$, if for any $y \in \mathrm{range}(R_1)$ and any $\gamma \in (0, 1)$ there are at most two distinct $x_1, x_2$ such that $f(y + \gamma x_1) - \gamma f(x_1) = f(y + \gamma x_2) - \gamma f(x_2)$ then there is no reward $R_2$ such that $G_2(\xi) = f(G_1(\xi))$ for all $\xi$.*

*Proof.* Suppose for contradiction that $G_2(\xi) = f(G_1(\xi))$ for all $\xi$. Let $\langle s, a, s' \rangle$ be an arbitrary transition. Applying Lemma 2, we get that

$$R_2(s, a, s') = f(R_1(s, a, s') + \gamma G_1(\zeta)) - \gamma f(G_1(\zeta))$$

for all trajectories $\zeta$ starting in $s'$. For clarity, let $x = G_1(\zeta)$ and $y = R_1(s, a, s')$, so that $f(y + \gamma x) - \gamma f(x)$. By assumption, there can be at most two distinct values $x_1, x_2$ such that $f(y + \gamma x_1) - \gamma f(x_1) = f(y + \gamma x_2) - \gamma f(x_2)$. However, Lemma 1 implies that there are at least three $\zeta_1, \zeta_2, \zeta_3$ starting in $s'$ with distinct values of $G_1$. Since $f$ is injective on $\mathrm{range}(G_1)$, this means that there are at least three distinct values of $x$ for which $f(y + \gamma x) - \gamma f(x)$ must be constant (and equal to $R_2(s, a, s')$), which is a contradiction. $\qquad\square$

## B  Towards Necessary and Sufficient Conditions

In this paper, we have provided several examples of "natural" policy orderings which cannot be represented using a reward function. It would be desirable to have a set of necessary and sufficient conditions to characterise those orderings over $\Pi$ that *can* be expressed by reward functions, similar to that provided by the VNM axioms (the VNM axioms themselves do not provide this, see Appendix C). We consider this to be an important topic for future work. In this section, we will discuss a few interesting properties which are shared by all policy orderings which can be represented by reward functions. We believe that these examples will help with building an intuition for what reward functions can and cannot express.

We would first like to point out that, while it seems difficult to characterise the *policy orderings* which can be expressed by reward functions, it is fairly straightforward to exactly characterise the sets of policies $\hat{\Pi}$ that can be *optimal* under some reward function:

**Proposition 1.** *A set of policies $\hat{\Pi}$ is the optimal policy set for some reward function if and only if there is a function $o : \mathcal{S} \to \mathcal{P}(\mathcal{A}) \setminus \varnothing$ that maps each state to a (non-empty) set of "optimal actions", and $\pi \in \hat{\Pi}$ if and only if $\mathrm{supp}(\pi(s)) \subseteq o(s)$.*

*Proof.* For the "if" part, consider the reward function $R$ where $R(s, a, s') = 0$ if $a \in o(s)$, and $R(s, a, s') = -1$ otherwise. The "only if" part follows from the fact that the optimal $Q$-function $Q^\star$ is the same for all optimal policies, so we can let $o(s) = \arg\max_a Q^\star(s, a)$. $\qquad\square$

This immediately lets us rule out many policy orderings as inexpressible. For example, consider the task "always go in the same direction" — this task cannot be expressed as a reward function, because any policy that mixes the actions of two other optimal policies must itself be optimal. It also shows that Markovian reward functions cannot be used to encourage *stochastic* policies. For example, there is no Markovian reward function under which "play rock, paper, and scissors with equal probability" is the unique optimal policy.

The next thing we would like to point out is that no reward function can express an ordering over $\Pi$ that has a *countable* number of equivalence classes (except trivial reward functions, which have only one equivalence class). This simple fact also rules out many orderings.

**Proposition 2.** *If $R$ is non-trivial then $J$ has an uncountable number of equivalence classes.*

*Proof.* This follows from the intermediate value theorem, and the fact that $J$ is continuous in $\Pi$. $\quad\square$

This simple observation can be used to e.g. create an alternative proof of Theorem 4, which says that the MaxSat objective cannot be represented as a (scalar) reward function. It also shows that objectives such as e.g. $J(\pi) = \min_{\xi \in \mathrm{supp}(\pi)} G(\xi)$, which evaluates policies according to the worst trajectory in their support, cannot be represented (since any policy then has the same value as some deterministic policy, and since there is only a finite number of deterministic policies).

## C  A Digression on the Von Neumann–Morgenstern Axioms

The famous VNM axioms, due to von Neumann & Morgenstern (1947), provide necessary and sufficient conditions for when a utility function can be used to represent a preference ordering for lotteries over a finite choice set. In an MDP, a policy induces a distribution over trajectories, and a reward function assigns a value to each trajectory. One might then wonder if the VNM axioms could provide necessary and sufficient conditions for when an ordering over $\Pi$ can be realised using

a reward function. This is not the case, and in this appendix, we briefly point out why. These results are not novel to this paper, but are instead provided to help with intuition building.

First of all, the VNM theorem assumes that the choice set is finite, whereas in an MDP, the number of trajectories is (countably) infinite. There are preferences between distributions over countable choice sets which satisfy the VNM axioms, but which can nonetheless not be represented using utility functions.[1] Second, not all distributions over trajectories can be represented as a policy (unless we allow both the policy and the transition function to be non-stationary). Third, there is a special structure to how a reward function assigns value to a trajectory, and not all functions $\Xi \to \mathbb{R}$ can be represented in this way. This means that the VNM axioms are not applicable to RL. However, it may still be possible to provide similar intuitive necessary and sufficient conditions for the RL case. We consider this to be an important topic for future work.

## D   MORE MORL OBJECTIVES

In this Appendix, we give even more examples of MORL objectives, and some comments on how to construct them – the purpose of this is mainly just to show how rich this space is. First, similar to the MaxMin objective, we might want to judge a policy according to its *best* performance:

**Definition 9.** *Given* $J_1 \ldots J_k$, *the* **MaxMax** *objective* $\prec_{Max}$ *is given by* $\pi_1 \ \prec_{Max} \ \pi_2 \iff$ $\max_i J_i(\pi_1) < \max_i J_i(\pi_2)$.

We would next like to point out that it is possible to create smooth versions of almost any MORL objective. In Section 5, we outline an approach for learning any continuous, differentiable MORL objective, so this is quite useful. We begin with a soft version of the MaxMax objective:

**Definition 10.** *Given* $J_1 \ldots J_k$ *and* $\alpha > 0$, *the* **Soft MaxMax** *objective* $\prec_{MaxSoft}$ *is given by*

$$J_{MaxSoft}(\pi) = \left( \sum_{i=1}^{k} J_i(\pi) e^{\alpha J_i(\pi)} \right) \Big/ \left( \sum_{i=1}^{k} e^{\alpha J_i(\pi)} \right).$$

This is of course not the only way to continuously approximate MaxMax, it is just an example of one way of doing it. Here $\alpha$ controls how "sharp" the approximation is – the larger $\alpha$ is, the closer $J_{\texttt{MaxSoft}}$ gets to the sharp max function, and the smaller $\alpha$ is, the closer it gets to the arithmetic mean function (so by varying $\alpha$, we can continuously interpolate between them). Similarly, we can also create a smooth version of MaxMin:

**Definition 11.** *Given* $J_1 \ldots J_k$ *and* $\alpha > 0$, *the* **Soft MaxMin** *objective* $\prec_{MinSoft}$ *is given by*

$$J_{MinSoft}(\pi) = \left( \sum_{i=1}^{k} J_i(\pi) e^{-\alpha J_i(\pi)} \right) \Big/ \left( \sum_{i=1}^{k} e^{-\alpha J_i(\pi)} \right).$$

As before, the larger $\alpha$ is, the closer $J_{\texttt{MinSoft}}$ gets to the sharp min function, and the smaller $\alpha$ is, the closer it gets to the arithmetic mean function We can also smoothen MaxSat:

**Definition 12.** *Given* $J_1 \ldots J_k$, $c_1 \ldots c_k$, *and* $\alpha > 0$, *the* **Soft MaxSat** *objective* $\prec_{SatSoft}$ *is*

$$J_{SatSoft}(\pi) = \sum_{i=1}^{k} \left( \frac{1}{1 + e^{-\alpha(J_i(\pi) - c_i)}} \right).$$

The larger $\alpha$ is, the closer $J_{\texttt{SatSoft}}$ gets to the sharp MaxSat function (and the smaller $\alpha$ gets, the closer $J_{\texttt{SatSoft}}$ gets to a flat $0.5$). And, again, this is of course not the only way to create a smooth version of MaxSat. It is unclear if it is possible to create a smooth version of ConSat without having any prior knowledge of (a lower bound of) the value of $\min_\pi J_1(\pi)$, but with this value it should be reasonably straightforward (see the construction in Theorem 5). As for LexMax, we can of course create a smooth approximation of it by taking a linear approximation of the weights, but here we would need some prior knowledge of $\max_\pi J_1(\pi) \ldots \max_\pi J_k(\pi)$.

---

[1]For example, consider the ordering that prefers all distributions with infinite support over all distributions with finite support, and which is indifferent between any two distributions in either of these classes.

# E  A METHOD FOR SOLVING MODAL TASKS

In this Appendix, we give an outline of one possible method for solving modal tasks. We mainly want to show that it is *feasible* to learn modal tasks, and so we only provide a solution sketch; the task of *implementing* and *evaluating* this method is something we leave as a topic for future work.

We will first define a restricted class of modal tasks, which is both very expressive, and also more amenable to learning than the more general version given in Definition 7:

**Definition 13.** *An* affordance *consists of a reward function and a discount factor,* $\langle R, \gamma \rangle$*, and an* affordance-based reward *is a function* $R^\diamond : \mathcal{S} \times \mathcal{A} \times \mathcal{S} \times \mathbb{R}^{2k} \to \mathbb{R}$*, that is continuous in the last* $2k$ *arguments. An* affordance-based MDP *is a tuple* $\langle \mathcal{S}, \mathcal{A}, \tau, \mu_0, R^\diamond, \gamma, \langle R, \gamma \rangle^k \rangle$*, where the reward given for transitioning from* $s$ *to* $s'$ *via* $a$ *is* $R^\diamond(s, a, s', V_1^\star(s) \ldots V_k^\star(s), V_1^\star(s') \ldots V_k^\star(s'))$*, where* $V_i^\star$ *is the optimal value function of the* $i$*'th affordance.*

This definition requires some explanation. In psychology (and other fields, such as user interface design), an affordance is, roughly, a perceived possible action, or a perceived way to use an object. For example, if you see a button, then the fact that you can *press* that button, and expect something to happen, is part of *how you perceive* it, in a way that might not be the case if you could somehow show the button to a premodern human. It can also be used to refer to a choice or action that is perceived as available in some context (without being tied to an object). Here, we are using it to refer to a *task* that could be performed in an MDP. The intuition is that $R^\diamond$ is allowed to depend on what *could be done* from $s$ and $s'$, in addition to the state features of $s$ and $s'$.

Before outlining an algorithm, let us first give a few examples of how to formalise modal tasks within this framework. First consider the instruction "you should always be able to return to the start state". We can formalise this using a reward function $R_1$ that gives 1 reward if the start state is entered, and 0 otherwise, and pair it up with a discount parameter $\gamma$ that is very close to 1. We could then set $R^\diamond$ to, for example, $R^\diamond(s, a, s', V_1^\star(s), V_1^\star(s')) = R(s, a, s') \cdot \tanh(V_1^\star(s'))$, where $R$ describes some base task. In this way, no reward is given if the start state cannot be reached from $s'$. Next, consider the instruction "never enter a state from which it is possible to quickly enter an unsafe state". To formalise this, let $R_1$ give 1 reward if an unsafe state is entered, and 0 otherwise, and let $\gamma$ correspond to a very high discount rate (e.g. 0.7). We could then set $R^\diamond$ to, for example, $R^\diamond(s, a, s', V_1^\star(s), V_1^\star(s')) = R(s, a, s') - V_1^\star(s')$, where $R$ again describes some base task.

These examples show that our "affordance-based" MDPs are quite flexible, and that they should be able to formalise many natural modal tasks in a satisfactory way, including most of our motivating examples.[2] However, the definition could of course be made more general. For example, we could allow the affordances to themselves be based on affordance-based reward functions, etc. However, it is not clear if this would bring much benefit in practice.

Let us now outline an approach for solving affordance-based MDPs using reinforcement learning, specifically using an action-value method. First, let the agent maintain $k + 1$ $Q$-functions, $Q^\diamond, Q_1, \ldots, Q_k$, one for $R^\diamond$ and one for each affordance $\langle R_i, \gamma_i \rangle$. Next, we suppose that the agent updates each of $Q_1, \ldots, Q_k$ using an off-policy update rule, such as $Q$-learning; this will ensure that $Q_1, \ldots, Q_k$ converge to their true values (i.e. to $Q_1^\star \ldots Q_k^\star$), as long as the agent explores infinitely often. Note that the use of an off-policy update rule is crucial. Next, let the agent update $Q^\diamond$ as if it were an ordinary Markovian reward function, using the reward $\hat{R}(s, a, s') = R^\diamond(s, a, s', V_1(s) \ldots V_k(s), V_1(s') \ldots V_k(s'))$, where $V_i(s)$ is given by $\max_a Q_i(s, a)$. In other words, we let it update $Q^\diamond$ using an *estimate* of the true value of $R^\diamond$, expressed in terms of its current estimates of $V_1^\star \ldots V_k^\star$. The fact that $Q_1, \ldots, Q_k$ converge to $Q_1^\star, \ldots, Q_k^\star$, and the fact that $R^\diamond$ is continuous in its value function arguments, will ensure that the estimate $\hat{R}$ also converges to the true value of $R^\diamond$. The update rule used for $Q^\diamond$ could be either on-policy or off-policy. We then suppose that the agent selects its actions by applying a Bandit algorithm to $Q^\diamond$, and that this Bandit algorithm is greedy in the limit, but also explores infinitely often, as usual.

This algorithm should be able to learn to optimise the reward in any affordance-based MDP. In the tabular case, it should be possible (and reasonably straightforward) to prove that it always converges to an optimal policy (assuming that appropriate learning rates are used, etc), using Lemma 1 in

---

[2]This arguably excludes "you should never enter a state where you would be unable to receive a feedback signal". However, this instruction only makes sense in a multi-agent setting.

Singh et al. (2000). We would also expect it to perform well in practice, when used with function approximators (such as neural networks). However, we leave the task of implementing and properly evaluating this approach as a topic for future work.

There are also several ways that this algorithm could be tweaked or improved. For example, the algorithm we have described is an action-value algorithm, but the same approach could of course be used to make an actor-critic algorithm instead. We also suspect that there could be interesting modifications one could make to the exploration strategy of the algorithm. If a standard Bandit algorithm (such as $\epsilon$-greedy) is used, then the agent will mostly take actions that are optimal under its current estimate of $Q^\diamond$. In the ordinary case, this is good, because it leads the agent to spend more time in the parts of the MDP that are relevant for maximising the reward. However, in this case, there is a worry that it could lead the agent to neglect the parts of the (affordance-based) MDP that are relevant for learning more about $V_1^\star \dots V_k^\star$, which might slow down the learning. Again, we leave such developments for future work, since our aim here only is to show that it is feasible to learn non-trivial modal tasks.

We also want to point out that the work by Wang et al. (2020) could provide another starting point for learning modal tasks using RL. In their work, they present some RL-based methods for determining whether a specification in Probabilistic Computational Tree Logic (PCTL) holds in an MDP. PCTL can be used to specify many kinds of properties of states in MDPs which depend on the transition function, including e.g. what states can and cannot be reached from a particular state, and with what probability, etc. We can therefore specify non-trivial modal tasks by providing a number of PCTL formulas, and allowing the reward function to depend on the truth values of these formulas. That is, we could consider a setup that is analogous to that which we give in Definition 13, but where the "affordances" are replaced by PCTL formulas. It should then be possible to learn tasks specified in this manner by using the techniques of Wang et al. (2020) to learn the values of the PCTL formulas, and then using ordinary RL to train on the resulting reward function.

