# OpenReview forum: "The Reward Hypothesis is False"
_ICLR.cc/2023/Conference — Submitted to ICLR 2023_

### Official Review · Reviewer_abfw · 2022-10-17

**Confidence:** 4
**Correctness:** 3
**Technical Novelty And Significance:** 2
**Empirical Novelty And Significance:** Not applicable
**Recommendation:** 3

**Clarity, Quality, Novelty And Reproducibility:**

The paper is quite clear in a technical sense, but story-wise it is quite unfocused. The paper is fairly polished. As outlined in another part of this review, I feel the paper's novelty is quite unclear, since many things are intuitive but it does not become very clear what are the contributions. Reproducibility is not a real issue, since the formalizations are purely theoretical and no experiments are provided.

**Strength And Weaknesses:**

The main strength of the paper is the topic, which receives a lot of attention in the last few years: reward functions, what can they do, what is their expressivity, how can they be learned/engineered/constructed, model/problem classes, and how to employ them in typical RL frameworks.

However, the paper does not really deliver on its main promises stated early in the paper (and the title). I guess it is fairly well known that, in principle, many situations exist where reward functions are multi-dimensional, or dependend on time or previous states, or something else, and then it is not possible to reduce it simply to a single scalar. However, for many (all?) problems it can be done, but typically with an exponential blowup in the parameters of the RL problem. For example, in the sub-area of non-Markovian reward functions, which is highly related to the "modal setting" in Section 4, problems with highly complex, non-Markovian reward functions can systematically be "compiled" into regular (single scalar, Markovian) problems by compiling necessary sequences of states (and actions) into a new state space (which is typically exponentially larger than the original). In this case the new problem can be solved using standard methods in deep RL, without any loss in accuracy or the possibility of finding an optimal solution, at a cost of a much increased state space, and accompanying measures like need for generalization, shaping rewards, longer learning times, and so on. Similar things can be said about the other two frameworks treated in the paper.

The main weakness of the paper is that it is highly unclear what the contributions are. The paper talks about well-known frameworks, and connects it more or less to RL, but without writing about what are contributions, new insights, and connections to the literature. For RL experts, I think most of it is very intuitive to read, and even interesting in the more general sections 1 and 5, but for me it is unclear how this paper advances the state-of-the-art and understanding of the reward hypothesis (relative to the literature). Furthermore, Sections 3 and 4 are even less worked out than Section 2, and only contain some initial thoughts. The modal RL is only sketched briefly, and does not connect well to the growing literature on combinations of RL and formal verification tools (like PCTL and temporal formalisms). Section 2 should first show the state-of-the-art in MORL, and then do additional things. Overall, all the methods are not illustrated nor motivated well, and also domain descriptions (examples) nor experimental evaluations or illustrations are incorporated. The paper as a whole has interesting bits and fragments, but as a paper, in this time and sub-area, it does not do well as "a research paper" since it is too unfocused and to unclear about its achievements.

Some other things:
- Since the paper is quite unfocused, I feel the related work is too. It is unclear why exactly this set of references is chosen.
- The preliminaries are ok, and formalisations too, but some "operations research" style formalizations are chosen (G,J,..) and it is good to connect (maybe) more literally to Sutton and Barto who have a more AI-oriented flavor of notation.
- Why not illustrate Sections 3 and 4 more with AI-oriented examples, in well-known domains, or even with some experimental part? This would help a lot in motivating these formalizations. Page 9 says "we have given many examples" but I feel this is really missing.
- The modal setting should be much more connected to the current trend to combine RL with verification tools, and it should get more space (maybe a separate paper) to really explain the setting in full detail.
- Section 5 should at least discuss in more detail how typically one can still "compile" problems to single-scalar problems, with the necessary trade-offs in model size, solution times, and so on.
- Too much detail is deferred to appendices, especially on page 9. The major part of the story should be in the paper itself though, and appendices should provide some further detail. The balance is off here.

**Summary Of The Paper:**

This paper is about the reward hypothesis (roughly that a reinforcement learning agent would only need one scalar signal to learn "any" task) and wants to show that this is not the case for many problems. The main portion of the paper consists of three settings (multi-objective, risk-averse and modal) and in each of these settings some formalizations and some properties are defined (and somewhat proven). The paper is purely theoretical.


**Summary Of The Review:**

This paper has several interesting things to say, about a topic that is currently active, but it is unclear what are the contributions, and furthermore it tries to do too many things in one paper without (almost) any illustrations, nor experiments, and overall the main portions sound not too novel at all, given the current state of the art in RL.

---

> ### Author Response · Authors · 2022-11-12
> **Response**
>
> We thank reviewer abfw for their time and feedback.
>
> We describe in great detail what are the exact contributions of our work, why they are novel, and how they extend existing work, especially in the Introduction, and in the Related Work section. We urge the reviewer to reconsider their impression that the main portions of this paper are "not too novel" -- all of our results extend the current literature, and are very relevant to current research directions.
>
> The issue of compiling tasks into Markovian reward functions has been brought up by several reviewers, and thus we have decided to respond to it in a longer, separate comment, see above. Note especially that the modal setting cannot be reduced to ordinary RL, even with an increase in the state space.
>
> We do explain what are the contributions, insights, and connections to the existing literature, see in particular the Existing Work section, the Introduction, and the Discussion. What we call "modal" RL is quite different from combinations of RL and formal verification -- if you feel differently, please point out to us what connections we have missed. As for Section 2; the contribution is to show when, and how, MORL can be reduced to scalar RL. This has not been demonstrated before. What, precisely, about the state-of-the-art in MORL do you think that we have unjustly left out? As far as we can tell, all relevant citations have been provided. Moreover, given the nature of our results, there are no particular experiments that we think would be relevant for this paper. What experiments do you think we should have included?
>
> We respond to the reviewers questions below:
>
> 1. What references do you feel that we should have included, or removed?
> 2. G and J, etc, are used by Sutton & Barto; see eg the "Summary of Notation" in their 2nd edition.
> 3. What experiments do you feel would elucidate the points made in this paper? The examples are given in the form of formalisations and proofs.
> 4. What connections between modal RL and verification do you feel that we should have explained better?
> 5. As we show in Sections 2-4, this cannot always be done. See also our general response to all reviewers above.
> 6. We are of course keen to improve this aspect of the paper, if you have specific suggestions. What do you think should be moved from the appendices to the main text?

---

> > ### Comment · Reviewer_abfw · 2022-11-15
> > **Thanks and some replies**
> >
> > Thanks for the replies. Maybe I should repeat that I do like the topic of the paper, and I certainly do not think the content is wrong. But that I do not think the paper is "coherent" enough to be one paper, and that, for me, there are various things unclear about the novelty and how it is situated exactly in "what we know about the possible characterizations of scalar reward functions" (my informal phrasing). I am on the reject-side of the spectrum for this particular version, but how far is less important (maybe I could have given a 5). I do believe that the re-phrasing of the start of the paper (including the title) makes much more sense.
> >
> > My main concern with the novelty starts with Section 2 about MORL. I know parts of the history of MORL through several papers in a long period, and there are several recent overviews too, but I am not up to date with very recent work. The novelty is argued by saying that (for example) Vamplev et al. approach the problem from a different angle (basically arguing that MO is natural for many aspects) but that they do not show that it is necessary, given that many problems cannot be reduced to a similar problem with the same policy preferences. Given the amount of work on solution algorithms, LP formulations, Pareto front analyses, and so on, I find it difficult to evaluate whether Theorems 2-5 are novel, also because they are not explicitly related (as far as I can see) to previous results (the only sentence saying something is "We greatly extend their work... significantly stronger" in Section 1), but how? I believe the authors if they say that their "mathematical approach" is novel, but from the main text (excluding appendices here) I can't see. And since the related work section also contains a remark "never... to the best of our knowledge" about modal RL, even though there it has been, I am unsure about the MORL related work too. Something similar holds for Section 3: I do not know much explicitly about risk-sensitive RL, but from the paper I do not get much to hold on to how this advances the state-of-the-art. Both Sections 2 and 3 do not contain references to other work/results, as far as I can see.
> >
> > There is a large literature forming around ideas to combine richer models of dynamical systems (with at its core versions of MDPs) specified by logics such as CTL, LTL, etc. with ideas from the model checking community, epistemic logic, etc. Temporal logics are modal logics with temporal modalities, but naturally one can go for knowledge/belief-like modal operators in a similar way. Extensions in the area of epistemic planning, or epistemic versions of Golog (based on situation calculus, used for various forms of MDPs too) can all be used to define MDPs with very complex goals and reward structures. The modal setting in section 4 is related to all of them. The mentioned reward-machines-approach is also related, but represents a specific automata-based approach that is more akin to hierarchical RL. Maybe just to give two specific references that represent a couple of developments more concretely to MDPs, and have given rise to several followups, and there are dozens of related papers in these areas, see also references in these papers. It is also related to the mentioning of "safety constraints" and "side effects".
> > Alshiekh, Mohammed, et al. "Safe reinforcement learning via shielding." Proceedings of the AAAI Conference on Artificial Intelligence. Vol. 32. No. 1. 2018.
> > Brafman, Ronen I., and Giuseppe De Giacomo. "Regular Decision Processes: A Model for Non-Markovian Domains." IJCAI. 2019.
> > Branching trees or not, CTL, LTL, etc.
> >
> > About the compilation of models; I think a huge distinction that should be made, is about finite/infinite domains (and also branching vs linear time structures in logics). If the underlying semantics has finite size, it seems natural to think that for a subset of these problems, the semantics of reward structures stays finite, and these problems could (in principle) be compiled with (huge) exponential blowups. The main issue, ofcourse, are for example infinite paths referred to by path expressions or expressions requiring logical omniscience over infinite structures (for epistemic operators) and so on. A discussion of these aspects would be interesting for a complete separate paper.
> >
> > Experiments are not "necessary" for this paper.

---

> > > ### Author Response · Authors · 2022-11-16
> > > **Responses (1)**
> > >
> > > Thank you for the reply! We had to split our responses into two comments, the first is below:
> > >
> > > Whether or not the paper is "coherent" enough is largely an aesthetic judgement, so it is difficult for us to give a strong response to this objection. All results in the paper are (partial) answers to the question; what things are impossible to express using Markovian reward functions? We think this is a natural question, and we do not think it is bad for the paper to cover many different topics and settings, because there are many different things that Markovian reward functions cannot express.
> > >
> > > The main existing work, which studies the expressivity of Markovian rewards in a mathematical way, is the work by Abel et al, 2021. We discuss their paper in both the 'Related Work'-section, and in our comment above. Our results are a substantive extension over their results.
> > >
> > > There is a huge amount of work in game theory and decision theory on what kinds of preference structures can and cannot be expressed using utility functions. This work does not directly carry over to the MDP setting, for the reasons we discuss in Appendix C.
> > >
> > > We have surveyed the existing literature to the best of our abilities, and have not found any of our results presented in any previous work. We can of course not entirely rule out the possibility that we have missed something, but we do not think we have. For example, if our results were present in the existing literature, then they would probably have been cited by Abel et al, 2021, or Vamplew et al, 2022, etc. If you think otherwise, is there any work you could point us to? If not, then it is difficult for us to reply to this point any more than we already have.
> > >
> > > (As a more minor point, we should also mention that we consider Theorem 1 to be the main contribution of Section 2. Theorems 2-5 are essentially just examples of how to apply Theorem 1; they could potentially have been labelled as corollaries rather than theorems. We even mention an alternative way to prove Theorem 4 in Appendix B. Therefore, even if one of Theorems 2-5 were to be proven in some earlier work, we would not consider that to reduce the contribution very much.)
> > >
> > > References to other work relevant to Section 2 and 3 are given in the "Related Work" section, and (to some extent) in Section 5, rather than directly in Section 2 or 3.

---

> > > ### Author Response · Authors · 2022-11-16
> > > **Responses (2)**
> > >
> > > Thank you for the reply! We had to split our responses into two comments, the second is below:
> > >
> > > We are aware of the literature on combining MDPs with ideas from automata theory, logic, and model checking, etc. We cite some of this work; Hasanbeig et al, 2020, Icarte et al, 2020, Wang et al, 2020, and Hammond et al, 2021. We also mention the connection to CTL in Section 4, and have a somewhat longer discussion on the connections to CTL and PCTL at the end of Appendix E. The reason why we most often mention the reward machines of Icarte et al is that many other techniques in this area compile the MDP into a structure that can be viewed as a special case of a reward machine. We can certainly include more references to this literature.
> > >
> > > We would, however, like to mention that we in this paper use the term "modal" in a somewhat more narrow sense than the sense of "modal logic". In particular, we use it to mean "pertaining to what is possible or impossible", as in eg https://plato.stanford.edu/entries/modality-varieties/. In that sense, LTL does not express modal statements, even though it is a modal logic, because LTL can only make assertions about what in fact occurs. CTL, on the other hand, can express modal statements (in the sense we use it), because it can make assertions about what could or could not occur. For that reason, not everything that relates to modal logic will be related to the setting we discuss in Section 4. To be more precise, the type of possibility we discuss is specifically "possibility according to the transition function"; we suppose this modality is most closely related to the nomic modality, or perhaps the metaphysical modality. This means that most modal logics (including eg epistemic logic, deontic logic, and doxastic logic, etc) are concerned with a different kind of modality than the modality we discuss in Section 4. Indeed, CTL and its extensions (PCTL, CTL*, etc) are probably the main example of modal logics that relate directly to the modal setting we discuss in Section 4. This connection is mentioned, and the main reference (Wang et al, 2020) is provided. The reason why we have refrained from discussing all the full minutiae of this issue in the paper is that it is very much in the weeds, and also not directly related to our results. However, we can of course include this (though it would probably have to go in the appendix).
> > >
> > > On the topic of "compiling" the problems by lifting the MDP state space; we agree that this, at least in some cases, can solve some of the problems we discuss. For example, in a finite-horizon problem, we can express the risk-sensitive objectives we discuss in Section 3 by including the time step in the state. However, this solution will not work for intinite-horizon problems (if we want the state-space to remain finite). Also note that this approach is unlikely to work for (most of) the problems we discuss in Sections 2 and 4. Lifting the state space is most likely to be useful when we want a solution that is non-stationary in the original state space. This is the case for the risk-sensitive problems we discuss in Section 3. However, it is not the case for many of the multi-objective problems we discuss in Section 2, or for the modal problems we discuss in Section 4 (see our response to point C4 by Reviewer UBdX). With the modal problems, the fundamental issue is that there are tasks where the reward that should be given for a trajectory depends on the transition function, whereas a reward function only can depend on the transitions in the trajectory. Lifting the state space can (presumably) not solve this problem, unless it is lifted in a way that depends on the transition function. However, we should also emphasise that this paper ultimately is about MDPs. Therefore, while we agree that a more complete analysis of these questions would be interesting, we believe that they are outside the scope of this paper. Note also that we mention these extensions as a topic for further work at the end of the discussion.

---

### Official Review · Reviewer_UBdX · 2022-10-25

**Confidence:** 3
**Correctness:** 3
**Technical Novelty And Significance:** 2
**Empirical Novelty And Significance:** Not applicable
**Recommendation:** 5

**Clarity, Quality, Novelty And Reproducibility:**

The paper reads well, and the reported statements seems to be correct from a brief inspection. However, the main take presented by this paper is not novel.

**Strength And Weaknesses:**

*Strengths*
- Overall clarity. The paper is well-written and easy to follow;
- Originality (partial). The paper provides an original formulation of the so-called modal RL problem, which might deserve its own characterization, as well as interesting insights on multi-objective RL formulations, and when those are not harder than standard RL.

*Weaknesses*
- Novelty. The result that is presented as the major take of the work, i.e., that Markovian rewards cannot express all the possible notion of tasks, is not novel, as it was already substantiated in (Abel et al., 2021);
- Presentation. The paper includes several potentially interesting results that take a backseat with respect to the (unsurprising) refutation of the reward hypothesis;
- Motivation. The analysis includes some arbitrary choices that lack a clear motivation, such as total policy ordering as to express the notion of task.

*Main Concerns*

(C1) Presentation. I do not understand the choice of the authors of presenting this paper as a formal refutation of the reward hypothesis. Abel et al. (2021) already demonstrated that some notion of tasks cannot be expressed through a Markovian reward function. Whereas the authors are right to say that their analysis mostly rely on different arguments, the major take is neither surprising or novel.

(C2) Notion of Task. I do not think the reward hypothesis ever mentioned policy ordering in the definition of "goals and purposes". One could argue that the actual goal of the learning process is to find an optimal policy, and that the expressivity of the reward should be evaluated in terms of the optimal policies it can induce. Nonetheless, Abel et al. (2021) demonstrated that also this weaker notion of task (which they call set of admissible policies) cannot be always expressed through a Markovian reward. The refutation argument of this paper does not include this result instead.

*Other Comments*

(C3) Generalization to average reward and finite-horizon.
To my knowledge, both the results presented in this paper and (Abel et al., 2021) are limited to the discounted RL setting. Do the authors believe that similar results could be generalized to the average reward setting, i.e., infinite horizon but without discounting, and the episodic setting, in which the reward function is often non-stationary?

(C4) Sufficiency of Markovian deterministic policies.
While it is well-known that the class of deterministic Markovian policies are sufficient to optimize any Markovian reward in the discounted setting, it is not clear what policy class we should consider for the presented generalized RL settings (multi-objective, risk-averse, modal RL). Do they also admit an optimal deterministic Markovian policy?

(C5) A "universal" problem setting.
The paper shows that some notion of tasks cannot be expressed through a Markovian reward function. Do the authors believe it might exist a more general (and tractable) problem setting that include all of the presented notions of task? Perhaps the convex RL formulation (Hazan et al., 2019; Zhang et al., 2020), where the learning objective is expressed as a convex/concave function of the state-action distribution, can be an interesting candidate. Another is the vectorial reward formulation by (Cheung, 2019).

(C6) Modal RL.
I am not really familiar with the notion of modal tasks, but the modal RL seems an original and interesting formulation to me. Can the authors provide a deeper characterization of this problem formulation? What kind of tasks it can express? E.g., it clearly subsumes standard RL tasks, what about multi-objective tasks or other generalization? Are the three presented problem setting non-overlapping in terms of expressivity?

(C7) Choice of the problem settings.
The choice of the problem settings to present in against the reward hypothesis is kind of arbitrary, and it is not clear why they have been preferred over constrained RL, pure exploration (e.g., maximum state entropy), imitation learning (as divergence minimization) or any other generalization.

(C8) Related works.
I believe that some additional works could be at least mentioned in the paper, such as the convex RL formulation and vectorial reward formulation, which are interesting generalization of the Markovian reward, as well as other papers considering non-Markovian rewards (e.g., through reward machines, for which I am not an expert and I cannot provide valuable references).

Convex RL:
- Hazan et al., Provably efficient maximum entropy exploration, 2019;
- Zhang et al., Variational policy gradient method for reinforcement learning with general utilities, 2020;
- Zahavy et al., Reward is enough for convex MDPs, 2021;
- Geist et al., Concave utility reinforcement learning: The mean-field game viewpoint, 2022;
- Mutti et al., Challenging common assumptions in convex reinforcement learning, 2022.

Vectorial rewards:
- Cheung, Exploration-exploitation trade-off in reinforcement learning on online markov decision processes with global concave rewards, 2019;
- Cheung, Regret minimization for reinforcement learning with vectorial feedback and complex objectives, 2019.

**Summary Of The Paper:**

This paper studies the ability of Markovian rewards to express tasks in some generalized RL settings, namely multi-objective RL, risk-averse RL, and the so-called modal RL. The tasks are intended as total policy ordering (multi-objective RL, modal RL) or total trajectory ordering (risk-averse RL). The paper provides examples of tasks that cannot be expressed by a Markovian reward, which they claim is a refutation of the reward hypothesis. The paper include several ancillary results, such as providing necessary and sufficient conditions for when a multi-objective RL task can be expressed through a Markovian reward, and a formulation of the modal RL problem, in which the tasks are expressed in modal terms (a notion close to counterfactuals) instead of categorical terms.

**Summary Of The Review:**

This paper is an interesting read, and it includes several ancillary results that can have significant value. Nevertheless, I believe that this paper cannot be accepted in its current form, as the presentation is centered on a result that is not novel. I would suggest the authors to reformulate the paper putting less emphasis on the refutation of the reward hypothesis, and more on the actual original contribution. E.g., modal RL seems a very interesting idea to expand, and some interesting ideas about how to approach them are relegated to the Appendix. Also the characterization of the necessary and sufficient conditions for which multi-objective RL cannot be casted as a standard RL problem are interesting, and they could provide a valuable contribution on their own.

---

> ### Author Response · Authors · 2022-11-12
> **Response**
>
> We thank reviewer UBdX for their review, and for their many thoughtful and insightful comments!
>
> After some consideration, we agree that the presentation ought to be changed, to better reflect the results we present. As several reviewers commented on this, we have responded to it in a separate comment, see above.
>
> Our responses to the questions, concerns, and comments, are:
>
> C1. This comment is fair. As several reviewers commented on it, we have responded to it in a separate comment above.
>
> C2. This is correct. Since several reviewers have commented on our choice to focus on policy orderings, rather than optimal policies, we have decided to respond to this more extensively in a separate comment, see above.
>
> C3. The episodic setting with stationary rewards is a special case of the infinite horizon setting (with stationary rewards), so all our results apply to that setting as well. The average reward setting and the non-stationary setting are both interesting, and deserve their own analysis. Our results do not cover either of these settings, and we do not know to what extent our results generalise to those settings as well. Our suspicion is that the average-reward setting is likely to be similar to the discounted setting, but that the non-stationary setting is likely to have more new and interesting properties.
>
> C4. This is a good question. The modal RL setting (as we have formalized it) always has an optimal deterministic Markovian policy (because there is always an ordinary reward function that is contingently equivalent to any modal reward), whereas the risk-averse setting may require non-Markovian policies (because whether or not it is worth taking a gamble in this setting can depend on how much reward you have got in the past). For MORL problems, it depends. For example, LexMax always has an optimal deterministic Markovian policy , whereas eg MaxSat may require non-Markovian policies (again because the right decision in a given state may depend on how much reward you have got in the past).
>
> C5. We consider this to be a very important question, and it has been in the back of our minds while working on this problem! We certainly hope that there exists some more general problem setting which is both tractable, and expressive enough to encompass the problem classes we discuss (as well as all other interesting problem classes). In that case, identifying this problem setting would be very useful. The convex RL formulation, as well as the vectorial reward formulation, are both interesting candidates, as is eg the reward machines of Icarte et al, 2020. Exploring this is a very interesting topic for further work.
>
> C6. We give some discussion of the types of tasks that this formulation covers in Section 4, especially in paragraph 3, as well as in Appendix E. One simple example would be "you get 1 reward if you reach this goal state, and -1 reward if you ever enter a state from which you cannot reach the initial state". This reward depends on the transition function, because the transition function determines from which states you can reach the initial state. The modal setting does not subsume the multi-objective setting, because (as per Theorem 10), for any modal reward function R1 and any transition function, there is a (scalar) reward function R2 that is contingently equivalent to R1, and (as per Theorem 1), scalar reward functions cannot express most multi-objective tasks. Our response to C4 also tells us that the modal setting does not subsume the risk-averse setting, etc. We do not expect any of these three classes to subsume either of the other two.
>
> C7. This is true. We have chosen these problem settings because we have interesting results concerning these settings, and not necessarily because we believe these problem settings to be more important than other problem settings, such as the ones you mention.
>
> C8. Thank you for bringing these related works to our attention; we will include them in the related work section.

---

> > ### Comment · Reviewer_UBdX · 2022-11-19
> > **After Response**
> >
> > I want to thank the authors for their thorough replies. The effort they are dedicating in addressing reviewers' concerns and incorporating their suggestions is truly remarkable. I think the authors are making some good points in their replies, such as noting that a greater knowledge of what kind of problems can be actually expressed through MDPs is paramount, and other clarifications on policy classes and problem settings. However, I believe reviewers are expressing some motivated concerns about the paper as well, and the coherence of the contribution has been extensively questioned.
> >
> > While I am currently keeping my original score, I would really like to reward the authors' dedication, and I am reserving some additional time to look at the updated manuscript to make a final evaluation.

---

### Official Review · Reviewer_UtAh · 2022-10-28

**Confidence:** 3
**Correctness:** 3
**Technical Novelty And Significance:** 3
**Empirical Novelty And Significance:** Not applicable
**Recommendation:** 6

**Clarity, Quality, Novelty And Reproducibility:**

The paper is clear and well-written. As far as I know, it is novel in discussing MORL, risk-sensitive and modal problems in the context of the reward hypothesis.


**Strength And Weaknesses:**

Strengths:
1. The paper is very well written and easy to follow. The introduction follows a clear narrative that takes the reader through all the class of problems and considerations. The subsequent sections also follow a consistent structure. Despite being dense in theorems, it’s easy to navigate.
2. The paper offers mathematical insight into each theorem in plain language. This also highlights the significance and logical progression of different theorems in this work.
3. Beyond showing MDPs where the goal cannot be expressed as scalar values, the author also devises solutions for these problems in section 5 as well as the appendix (modal problems).


Weakness:
I believe that the paper would be stronger if the author could address the following questions. Because there is a wide range of interpretations of the reward hypothesis, I hereby divide my questions into “philosophical” and “technical” ones:

Philosophical:
1. Response by Sutton et al. The authors’ argument for refuting the reward hypothesis is based on the fact that not all “goals and purposes” can be represented by scalar reward. This reminds me of one of Rich Sutton’s responses that goals ultimately lead to some desired behavior/policies. And one can trivially design a reward function to describe such optimal policy by penalizing actions that are not in the behavior. I wonder if the authors disagree with this. This may also be a point worth clarifying in the paper.
2. It’s not that the hypothesis is False, it’s MDP’s fault. One of the main reasons that risk-averse learning and modal learning fail the reward hypothesis is because they contradict the Markov assumption. I.e. for risk-averse learning, the reward depends on the agents’ future actions and for modal learning, the reward depends on the transition function. Since the reward hypothesis, as cited, does not imply Markov conditions, I wonder if the authors should narrow the claim to “The reward hypothesis is false under the Markov assumption”?
3. Undesirable/unsatisfactory is not impossible. In modal learning, the author points out that constructing a scalar reward function for modal learning is “undesirable” as it requires an enumeration of all the states and actions and detailed knowledge of the environment. However, if I understand correctly, it is still “possible” and there exists such a scalar reward function in any specific env. Sutton et al noted this in their post saying “is it always beneficial to describe goals as rewards in practice? Probably not.” I believe this doesn’t refute the original reward hypothesis that constructing a reward function is possible. Please let me know if I didn’t understand this part correctly.

Technical:
1. Non-differentiable MORL problem. The authors mention in section 5 that MORL problems with differentiable U can be solved via existing solvers. However, if I understand correctly, MaxSat is a non-differentiable MORL problem. Do the authors have any suggestions for solving this problem?


**Summary Of The Paper:**

This paper discusses three types of problems where the goal cannot be expressed as scalar values in MDP (MORL where the objective cannot be expressed as weighted sums of individual objectives, trajectory-wise risk-sensitive problems, and modal problems). They also propose potential solutions for modal problems and discussed relevant literature for solving MORL and risk-sensitive problems.

**Summary Of The Review:**

I’m not yet convinced by the arguments about modal problems, and I also think that risk-sensitive problems are limited by the expressivity of MDP, not necessarily because the reward hypothesis is False. I am interested in hearing the author’s response, and would either revise my review or suggest that the authors adjust the claim of the paper to be less strong than the “reward hypothesis is False”.

---

> ### Author Response · Authors · 2022-11-12
> **Response**
>
> We thank reviewer UtAh for their thoughts and time.
>
> Our responses to your questions:
>
> Philosophical questions:
>
> 1. Yes, for any (deterministic, stationary) policy \pi, there is a Markovian reward function R such that \pi is the only policy that is optimal under R. This is absolutely worth clarifying! In the paper, we have decided to focus mainly on *policy ordering*, rather than *optimal policies*. Since several reviewers brought up this point, we have decided to give a more comprehensive response to it in a separate comment, see above.
> 2. After some consideration, we agree that it probably would be appropriate to narrow down the central claim of the paper. This was also highlighted by several reviewers, and so we have responded to this point in a separate comment, see above.
> 3. Yes, you are correct; reducing modal problems to an ordinary reward signal is still (theoretically) possible in each specific environment. The reason why we still consider this to be a point against the reward hypothesis essentially has to do with the order of the quantifiers; for each modal RL problem and each environment, there exists an ordinary reward function that encodes that problem in that environment. However, it is not the case that for each modal RL problem, there exists an ordinary reward function that encodes that problem in each environment. This means that we can reduce modal RL problems to an ordinary reward function if we know the environment, but not if we don't know the environment. We consider the latter case to be the more important one, since one of the main points of RL is that it is meant to be applicable in unknown environments. Note also that even in cases where the environment is known, constructing the corresponding reward function could be very difficult (indeed, the difficult may be comparable to the difficulty of solving the problem in the first place).
>
> Technical questions:
>
> 1. Yes, MaxSat is indeed a non-differentiable MORL problem. We do not have any concrete suggestions for how to solve this, except maybe to solve a differentiable approximation thereof (such as the one given in Definition 12 in Appendix D). We bring up MaxSat mainly for illustrative purposes, but designing an effective method for solving it could be an interesting direction for further work.

---

> > ### Comment · Reviewer_UtAh · 2022-12-07
> > **Updated my scores because the authors have addressed most of my concerns**
> >
> > I would like to thank the authors for their response. I appreciate that the authors have adjusted their claims and added additional discussions related to Philosophical questions (1)(2) and offered more clarifications on the technical question (1).
> >
> > I am still not fully convinced of the Philosophical question (3). I agree that it is important to be able to extend RL reward functions to unknown environments, and a reward function that requires knowledge of the environment is not practically useful. However, in principle, this doesn’t refute or challenge the original reward hypothesis. It just simply emphasizes that it’s not practical.
> >
> > Given that the authors have addressed the majority of my concerns, I have updated my scores.

---

### Official Review · Reviewer_dSf2 · 2022-10-29

**Confidence:** 4
**Correctness:** 3
**Technical Novelty And Significance:** 3
**Empirical Novelty And Significance:** Not applicable
**Recommendation:** 8

**Clarity, Quality, Novelty And Reproducibility:**

Clarity: Strong; given the technical nature of the results it was fairly easy for me to follow along.

Quality: Strong; the results are non-trivial and argue against a commonly-believed hypothesis and (to my knowledge) are correct (barring minor issues that can be easily corrected).

Originality / Novelty: While the results have been discussed informally before, to my knowledge this is the first time they have been formally proved.

Reproducibility: N/A, no experiments

**Strength And Weaknesses:**

Strengths:
1. The paper adds to the literature showing that the reward hypothesis is false (as previously demonstrated by Abel et al, which received an Outstanding Paper at NeurIPS, and discussed informally in the AI alignment community, e.g. [1]). To my knowledge, while the specific failures shown in this paper were generally suspected to be issues with the reward hypothesis, this is the first time they have been shown formally. I find the risk aversion result particularly strong.

Weaknesses:
1. In its discussion of MORL, the paper primarily analyzes policy orderings, even though what we typically care about in practice is the chosen policy. When considering the chosen policy, there is a significantly stronger argument that a linear combination of (history-based) rewards captures what we want (discussed below).
2. I found the “modal reward” section discussion to be fairly unconvincing, as it is not clear why we should care about robust equivalence rather than contingent equivalence (discussed further below).

**MORL**

I think there is a fairly strong argument for single-objective learning given _trajectory-based_ rewards, based on applications of decision theory to decision-procedures over trajectories (i.e. policies). Specifically:

1. We assume that when considering each $R_i(\xi)$ individually, we would like to choose trajectories on the basis of expected value under $R_i$. When S and A are finite, and we have a finite horizon length, this can be justified by the VNM theorem, since there are only a finite number of trajectories. Of course, this may require transformations to encode e.g. risk-aversion (such as switching from amount of money to log(amount of money), to be risk-averse w.r.t.money).
2. Similarly we assume that our final decision-procedure after aggregating should also choose trajectories on the basis of expected value under some reward function (once again, this is just saying that it should be VNM-rational).
3. We make the hopefully-uncontroversial assumption that if all of the $R_i$ are indifferent between two trajectories, then so too is our final decision-procedure.
4. The Harsanyi Aggregation Theorem [2] then says that our decision-procedure can be represented as maximizing the weighted linear sum of the various $R_i$.
5. The (history-dependent) policy that maximizes the weighted linear sum of the various $R_i$ is obviously consistent with this decision-procedure, and so is a good target to learn, which can be done with single-objective RL

(I have read Appendix C and I do not think this argument falls prey to any of the mistakes you mention there. Still, I have not checked the argument carefully; there could be some problem.)

Of course, this does not save the reward hypothesis (as you define it) because it requires us to use _trajectory-based_ rewards, whereas you operationalize the reward hypothesis as using _Markovian_ rewards. Nonetheless, I think it is worth mentioning this argument in favor of single-objective RL (if only in an appendix).

What happens if you try to run your argument on this construction? The first obstacle is that I am using trajectory-based rewards while your argument assumes Markovian rewards. However, I would guess that your argument applies just as well to trajectory-based rewards, as the key linearity-based arguments would still apply.

In that case, your result would say that for specific MORL objectives, the resulting policy ordering cannot be expressed by a single trajectory-based reward. One possibility is that the MORL objectives choose policies that are not consistent with a VNM-rational decision-procedure over trajectories, violating step 2 of my argument; if this were the case I would take that as a strike against MORL objectives rather than a strike against the reward hypothesis.

However, the more likely scenario is that both claims are simultaneously true. My argument only makes claims about the final decision-procedure over trajectories, i.e. the _optimal_ policy chosen by the single trajectory-based reward. It could indeed be the case that the ordering over other, non-optimal policies is different between the MORL objective and any possible trajectory-based reward. However in this case it is unclear why I should care. Ultimately we only obtain and deploy one policy; why should I care what the ordering over other policies is?

**Modal rewards**

It is not clear to me why we should ask for robust equivalence instead of contingent equivalence. As I understand it, your argument for this claim is:

> the construction of R♢τ will invariably be laborious, and require detailed knowledge of the environment. For example, consider the task “you should always be able to return to the start state”; here, constructing R♢τ would amount to manually enumerating all the states from which the start state is reachable. This is very much against the spirit of reinforcement learning, where much of the point is that we want to be able to specify tasks which can be pursued in unknown environments.

However, it is unclear to me how you do any better with a modal reward function: how do you write down R(s, a, s’, τ) in full generality? I’m not familiar with all of the references you cite, but in e.g. Krakovna et al., 2018; 2020a, Turner et al., 2020, and your own Appendix E, there is no R(s, a, s’, τ) that gets written down and then maximized; rather the algorithms work by automatically building the contingently-equivalent reward R(s, a, s’) by learning it from experience. Why should we not think of this as “writing down the contingently-equivalent reward, and then maximizing that”?

The one difference that I see is that the contingently-equivalent reward is built _at the same time_ that the policy is learned, but this is surely just an efficiency improvement: you could also design methods that first learned the contingently-equivalent reward, and then maximized it. Given that the reward hypothesis (as you have operationalized and analyzed it) is primarily about what is theoretically possible; an argument that depends entirely on an efficiency difference does not seem appropriate.

(You might also say that the fact that you have to learn the reward means it is very hard to provide in practice, which is a strike against the reward hypothesis, but in the introduction you emphasize that the reward hypothesis is about what can be done _in principle_ and not about what is difficult to do in practice.)

Minor issues:

Lemma 1 is false as stated: given some s’, the reward could be constant on the states reachable from s’, while being non-constant on states unreachable from s’. In the proof, the issue arises here:

> Finally, note that this means that we can construct such trajectories for any state s′, by simply composing a transition ⟨s′, a, s⟩ with each of ζ1, ζ2, ζ3.

This doesn’t work if that transition is impossible (as in the preliminaries you require trajectories to be “possible according to µ0 and τ”). (If you remove this requirement I would expect this breaks other proofs.)

[1] Ngo, Richard. “Coherent behaviour in the real world is an incoherent concept.”

[2] Harsanyi, John C. "Cardinal welfare, individualistic ethics, and interpersonal comparisons of utility." Journal of political economy 63.4 (1955): 309-321.

**Summary Of The Paper:**

This paper operationalizes the reward hypothesis as: Any task of interest can be captured as the maximization of the expected sum of a scalar reward signal R(s, a, s’). They then show several ways in which this hypothesis fails:

1. Many natural task formulations such as “Do X, but break ties according to Y” can be formulated using multi-objective reinforcement learning. The policy ordering induced by such a formulation cannot be expressed by any reward signal R(s, a, s’).
2. It is common to be risk-sensitive in pursuit of some goal: for example, if M() captures the amount of money you have, then you often care a lot if M() becomes very low or zero, but don’t care as much if M() increases by a similar amount. In economics this is often handled by transforming M() into some new function M’, such as M’(x) = -exp(αM(x)). In RL, we might similarly want to choose a risk-averse policy that corresponds to a Q function Q’(s, a) = -exp(αQ(s, a)). However, barring uninteresting edge cases, there is no reward function R’ that induces Q’, due to the sequential nature of the task and how R’ is _summed_ to get Q’.
3. We may want task specifications that depend on counterfactuals and possibilities, e.g. “accomplish X while remaining _able to reach_ the start state”. These can be formalized as reward functions that depend on the transition function. While it is of course possible to encode this as a regular reward function for a given transition function, there is no regular reward function that works for all transition functions (again barring uninteresting edge cases).

**Summary Of The Review:**

This paper provides clear technical results arguing against a commonly-held hypothesis. While I do have some nitpicks, they are just that: nitpicks.

---

> ### Author Response · Authors · 2022-11-12
> **Response**
>
> We thank reviewer dSf2 for their detailed comments and feedback.
>
> Our responses to the comments and questions are below, organised as in the review:
>
> 1. Since several reviewers commented on our decision to focus on policy ordering, rather than optimal policies, we have responded to this in a separate comment, see above.
> 2. Our point here essentially has to do with the quantifier order; for each modal RL problem and each environment, there exists an ordinary reward function that encodes that problem in that environment. However, it is not the case that for each modal RL problem, there exists an ordinary reward function that encodes that problem in each environment. This means that we can reduce modal RL problems to an ordinary reward function if we know the environment, but not if we don't know the environment. We consider the latter case to be the more important one, since one of the main points of RL is that it is meant to be applicable in unknown environments.
>
>
> MORL:
>
> Thank you for this argument, it was very interesting to read and consider.
>
> As far as we can tell, it should be possible to adapt our argument from Theorem 1 to this setting, with the conclusion being that some MORL objectives induce preference orderings over lotteries of trajectories that are not VNM compliant. However, this is perhaps not too surprising; lexicographic preferences are famously not VNM compliant, and neither are eg "satisficing" preferences, such as "ensure that you obtain at least X utility with probability at least p", etc.
>
> The issue of whether we should care about the policy order, or the optimal policy, was brought up by several reviewers, and so we decided to respond to it in a separate comment above. We would also like to note that your argument (as far as we can tell) in fact does make claims about the policy order, and not just the final decision procedure, since the VNM axioms concern all preferences between any pair of lotteries (and a lottery is akin to a policy).
>
>
> Modal rewards:
>
> Yes, our suggested approach for solving modal problems does build and maximise an ordinary reward function, and these steps could in theory be de-coupled. However, which reward function is built will be different in different environments -- this is why we think it is reasonable to say that the resulting reward function does not fully encode the task, in the way we typically want from a reward function. For example, if we undergo distributional shift, then we would have to re-engineer the reward function, which is not typically the case. There are also probably ways to solve modal tasks without building an intermediate reward function.
>
>
> Minor issues:
>
> Lemma 1 is not false as stated. In the paragraph above the lemma, we state:
>
> "In this section, we will consider the domain of $G$ to be the set of all coherent trajectories, \emph{not} the set of trajectories which are possible under some transition function $\tau$. The reason for this is that we do not want to presume any prior knowledge of the environment. If we restrict the set of trajectories we consider, then some risk-averse utility functions can become possible to express (consider the case of a tree-shaped MDP, for example)."
>
> This choice is only made for Section 3, but it does not matter much for Section 2 or 4, since they mainly work with policy orderings rather than trajectories. We could thus let this convention hold throughout the paper.

---

> > ### Comment · Reviewer_dSf2 · 2022-11-27
> > **Thanks for the response.**
> >
> > I mostly agree with this response.
> >
> > I'm still not particularly convinced about the modal rewards setting. I agree that the quantifier order matters, and I agree that distributional shift can change the contingent reward you build in practice, but neither of these seems to me to be particularly important for "what rewards can express in principle", which is the theme of the rest of the paper.
> >
> > In any case, I still vote for acceptance of the paper, given the presence of significant non-trivial technical results that should be of interest to the RL community.

---

### Official Review · Reviewer_p6jG · 2022-10-31

**Confidence:** 3
**Correctness:** 3
**Technical Novelty And Significance:** 3
**Empirical Novelty And Significance:** Not applicable
**Recommendation:** 6

**Clarity, Quality, Novelty And Reproducibility:**

The mathematical statements themselves were clearly defined and very readable. They were interesting results. Regarding originality, the authors have discussed the novel case of "Modal RL" which I have not seen elsewhere, and found very interesting. The other 2 objective functions as nonexpressible in an (S,A, reward) MDP were novel/curious demonstrations as well.

However, when the authors say in the Discussion as their main point "We argue that our results show that the reward hypothesis is false – there are tasks, which are natural to state and intuitive to understand, and which can be solved with RL methods, but which cannot be expressed using scalar Markovian reward functions", Abel et al 2021 has already investigated tasks that cannot be expressed using a single scalar reward and this (expressivity) was their main point. The authors should therefore state that their study adds to this previous demonstration.

**Strength And Weaknesses:**

The mathematical demonstrations are themselves interesting and curious in their own right. The main conclusions for each of the 3 individual objective functions appear to be justified.

However, their main purpose in this paper, as formulated by the authors, is to serve to disprove the "reward hypothesis" and this framing is the problematic part.

The first problem with the framing of in terms of the reward hypothesis is that in Sutton & Barto itself, this statement is explicitly stated to be an informal or guiding idea rather than a solid dictum. The second problem is that the reward hypothesis is broadly about the idea that goals can be formulated as rewards, and is therefore agnostic to the methods used to achieve this control problem or even the formulation of the setting itself (whether MDP or otherwise). Therefore, mathematical proofs about the nonequivalence of MDP RL and the 3 objective functions considered cannot disprove this statement (which itself, was informal to begin with), and framing this as a disproving is flashy but doesn't necessarily contribute to the advancement of this field.

It is not to say that the paper did not demonstrate some interesting things. Examining expressivity of singular reward functions/MDPs is a cool question, and within this more targetted focus, this paper brings some interesting new demonstrations.

All in all, the authors should either provide clarifications to rigorously justify their current framing (i.e. disproving) or, they can amend their whole manuscript angle to be closer to what the authors actually demonstrated -- i.e. that these other objective functions are nonexpressible in an (S,A, reward) MDP.

**Summary Of The Paper:**

This paper investigates tasks in which the "reward hypothesis" is not the most appropriate framework. The authors demonstrate three types of objectives: MORL,  risk sensitive RL, and Modal objectives, that cannot mathematically be reduced to a singular MDP formulation.

**Summary Of The Review:**


This study can be seen as an extension of the work Abel et al 2021, by providing 3 new and interesting tasks that scalar rewards cannot express.

With reorganization and rigorous framing, this paper can potentially be an interesting one to the community.

---

> ### Author Response · Authors · 2022-11-12
> **Response**
>
> We thank reviewer p6jG for their thoughts and feedback.
>
> We agree that our work can be seen as a continuation of the work by Abel et al, 2021. Note that we discuss this connection quite extensively in the first paragraph of the related work section, including the exact ways in which we extend and strengthen their results. We also comment on this more in our separate comment above.
>
> We acknowledge your concerns about our framing -- we give a more comprehensive response to this in our separate comment above.

---

> > ### Comment · Reviewer_p6jG · 2022-11-24
> > **Comment**
> >
> > Thanks to the authors for answering our concerns. Since the title and relevant sections have been amended to describe more directly the actual results that were found, I think this paper is both more clear in its message, and acceptable. I have altered my score to one grade above.
> >
> > As I previously said, the mathematical demonstrations are themselves interesting and curious in their own right.

---

### Official Review · Reviewer_Y3kp · 2022-11-04

**Confidence:** 2
**Correctness:** 3
**Technical Novelty And Significance:** 2
**Empirical Novelty And Significance:** Not applicable
**Recommendation:** 5

**Clarity, Quality, Novelty And Reproducibility:**

Clarity - The writing in general is followable and most of the proofs are straightforward.

Novelty - The attempt to disprove the reward hypothesis mathematically is novel.

Reproducability - It is reproducible


**Strength And Weaknesses:**

Strengths
The paper does provide sound proofs that some of the multi objective and risk averse based reward functions cannot be reduced, ceteris paribus, to an equivalent scalar Markovian reward function.

Weaknesses/Questions

1. The paper states that ‘In other words, it is the hypothesis that any natural task can be expressed as a reward signal’. The definition of a natural task can vary significantly. It would also bring up the question of whether the corresponding objective functions are ‘natural’ or if they are part of ‘what we mean by goals and purposes’.

2. I think there should be clarification on why the state space has to be the same between the original and modified MDPs to showcase that the reward hypothesis holds.

3.This also leads to the question of if there are scalar rewards conditioned on the objectives then would that be equivalent to the original MORL or Risk Sensitive RL reward functions or if there are reward function approximators that approximate such equivalent rewards then would the reward hypothesis hold. Could there even be such approximators?

4. Coming to modal reward function, there isn’t a specific example that showcases how the reward function might be dependent on the transition function.


**Summary Of The Paper:**

The paper tries to disprove the reward hypothesis mathematically by showing some classes of tasks cannot be expressed using any scalar Markovian reward function. They showcase a set of multi-objective reward functions, risk averse utility functions that can’t be reduced to an equivalent scalar reward function. They also introduce a new class of RL tasks, namely,  modal tasks where the reward is dependent on the transition function

**Summary Of The Review:**

The paper definitely provides more insight into discussing the reward hypothesis and the extent to which it can be flexible but there are other aspects that need to be clarified (mentioned in the strengths and weaknesses section) in order to firmly disprove the reward hypothesis. Even in regards to the modal tasks, additional details are needed to clarify the dependency of the reward function and the transition function. Due to these reasons, I would incline towards not accepting this paper.

---

> ### Author Response · Authors · 2022-11-12
> **Response**
>
>
> We thank reviewer Y3kp for their feedback. Our answers to their questions are:
>
> 1. The definition of a "natural" task can of course vary considerably. In this paper, our (implicit) operationalisation is, essentially, a task that one might reasonably want an RL system to solve, where a task is formalised as a policy ordering in an MDP, and "reasonably" is left undefined, but taken to apply to each of the three settings we discuss. We could unfortunately not understand what you meant by "whether the corresponding objective functions are ‘natural’ or if they are part of ‘what we mean by goals and purposes’" -- could you please clarify this for us?
> 2. Since several reviewers brought up this point, we have responded to it in a separate comment above.
> 3. We could unfortunately not quite understand this question -- could you please clarify it for us?
> 4. We discuss this in a somewhat loose way in the third paragraph of Section 4, and in a more formal way in Appendix E. One simple example would be "you get 1 reward if you reach this goal state, and -1 reward if you ever enter a state from which you cannot reach the initial state". This reward depends on the transition function, because the transition function determines from which states you can reach the initial state.
>
> See also our comment above about our proposed reframing of our results.

---

### Author Response · Authors · 2022-11-12
**Common Responses**

We would like to very much thank the reviewers for their thoughtful feedback to our manuscript!

Some points were brought up by multiple reviewers. Therefore, we have decided to respond to those points in one single comment here. In addition to this, we have of course also given more detailed responses to each individual review.

1.

Several reviewers commented on our choice to frame our results in terms of the reward hypothesis, though for somewhat different reasons. In particular, if we interpret the reviews correctly, some reviewers say that our results come short of refuting the reward hypothesis, whereas other reviewers say that the reward hypothesis has already been refuted by Abel et al, 2021. We will respond to both of these points; for simplicity, we will start with the second point.

In the Related Work section, we discuss the work by Abel et al, 2021, and how our work builds on theirs. In short, in relation to the reward hypothesis, Abel et al demonstrate the *existence* of tasks which cannot be formalised using (scalar, Markovian) reward functions (where a task is operationalised as either a set of optimal policies, or a policy ordering, or a trajectory ordering). However, the mere existence of such tasks does not tell us whether we should expect to encounter such tasks in practice. For example, for each of these operationalisations, there are tasks which are uncomputable. Such tasks can probably not be captured by (scalar, Markovian) reward functions, but this is not worrying, because in practice, we would never want to give such a task to an RL agent. On the other hand, we provide concrete examples of broad, useful classes of tasks which one might want an RL agent to complete, and demonstrate that they cannot be formalised using (scalar, Markovian) reward functions. We therefore extend the results by Abel et al, by showing that the limitations in the expressivity of (scalar, Markovian) reward functions is not merely a "mathematical" issue, but a practical issue as well. Stated differently, we provide a large amount of additional detail and insight into precisely what kinds of tasks we should expect (scalar, Markovian) reward functions to be unable to capture, and when we should expect to encounter such tasks in practice.

As for the other point (that our results do not refute the reward hypothesis), p6jG emphasises that the reward hypothesis is stated informally, and thus there is a sense in which mathematical results cannot disprove it as a whole, but only one particular formalisation of it. Relatedly, UtAh remarked on the fact that we have only disproven the reward hypothesis in the case of MDPs, but that it might hold in non-Markovian environments. Our response to this is that this objection is completely true: our analysis only concerns Markovian reward functions in Markovian environments without augmented state spaces. However, this is the setting in which almost all reinforcement learning is done, and methods for other settings (such as POMDPs and MOMDPs) often leverage methods designed for MDPs. We therefore believe that this is the most interesting setting to analyse.

This being said, the impression we get from the feedback of the reviewers is that our choice to frame our results in terms of the reward hypothesis may be unnecessarily divisive or distracting from the actual, formal results we provide. We would like to emphasise the following points:

 * It is important to know what can and cannot be formalised within a particular framework. In RL, the most common problem setting is MDPs. It is therefore important to know what kinds of tasks can and cannot be expressed in MDPs.
 * We provide several formal results about what kinds of tasks can and cannot be formalised in MDPs, which give a detailed, intuitive picture of some practical situations in which we should expect to be unable to express a problem as an MDP in a satisfactory way. These results extend the existing literature in several ways.
 * Even if some limitation in the expressiveness of MDPs can be amended by moving to a different setting (such as POMDPs, MOMDPs, or non-Markovian environments), it is still important to know when this is necessary, and when the MDP setting suffices.

This explains the context of our contribution. To make this more salient, we propose that we change the title of our paper, and the corresponding parts of the introduction and discussion, to better point to the results we provide. We have the following suggestions:

 * Three Problem Classes That MDPs Cannot Express
 * The Reward Hypothesis Is False in MDPs
 * Three Limitations in the Expressivity of Markov Rewards

We want to give our results the best possible framing, and are thus keen for your feedback on these suggestions.

---

> ### Author Response · Authors · 2022-11-12
> **More Common Responses**
>
> 2.
>
> Some reviewers have commented on our choice to formalise RL tasks in terms of a policy ordering, rather than as a set of optimal policies. As pointed out by Reviewer UtAh, it is well-known that for any (deterministic, stationary) policy \pi, there exists a reward function R such that \pi is the only policy that is optimal under R -- simply let R(s,a) = 0 if \pi(s) = a, and -1 otherwise. In that sense, any desired behaviour can be made the optimal solution to some reward function (provided that that behaviour is deterministic and stationary).
>
> However, we believe that the policy ordering is a better operationalisation of a "task" than is the optimal policy. In complex environments, we can typically not find the optimal policy, and instead, we aim for finding a policy that is as good as possible. However, this means that it is not enough for the reward function to have the right optimal policy; it must also induce the right preferences between the (sub-optimal) policies that the policy optimisation algorithm actually considers. The only way to robustly ensure this is if the reward function induces the right policy ordering.
>
> Another, less central, comment on this issue is that the reward function where R(s,a) = 0 if \pi(s) = a, and -1 otherwise is very hard to construct. Indeed, it is as hard to construct as the optimal policy itself, and if we have the optimal policy, the reward function is no longer needed. The way that a reward function is constructed in practice is typically more analogous to reasoning about the pairwise preferences between policies that the reward function is likely to induce.
>
> We therefore believe that the policy ordering is a more interesting formalisation of a "task" than are the optimal policies. We will amend the paper to make this point more clear.
>
>
> 3.
>
> Some reviewers commented on the possibility of alleviating the limitations in the expressivity of (scalar, Markovian) reward functions by augmenting the state space of the underlying MDP. We suppose that what they have in mind must be something analogous to the "reward machines" of Icarte et al, and similar constructions?
>
> In short; increasing the state space of the MDP can certainly increase the expressive power of (scalar, Markovian) reward functions, see e.g.\ Icarte et al, 2020. Whether or not this solves the problems we bring up in this paper is beyond the scope of our work. For risk-averse RL, we suspect that this does help, for modal RL, we suspect that it does not help, and for MORL, we are unsure. To determine either of these three cases conclusively, additional analysis is needed. We would also like to again emphasise that even if some limitation in the expressiveness of MDPs can be amended by moving to a different setting, it is still important to know when this is necessary, and when the MDP setting suffices. We will make this more clear in the main paper.
>
> In addition to these responses to some common objections, we have also given an individual response to each reviewer.

---

### Author Response · Authors · 2022-11-12
**Amended Version**

We have uploaded an amended version of the manuscript, which incorporates much of the feedback we have been given.

The main difference is that we have changed the title, abstract, intro, and discussion to put less emphasis on the reward hypothesis. We have also included a new section 1.3 where we discuss the choice to focus on policy orderings instead of optimal policies, and updated many other parts of the text in accordance with the suggestions from the reviewers.

---

> ### Comment · Reviewer_UBdX · 2022-11-19
> **Highlighting changes**
>
> Can the authors highlight the changes made to the manuscript in a different color, so that reviewers can check the new version without reading everything again?

---

> > ### Author Response · Authors · 2022-11-21
> > **Highlighting**
> >
> > Of course. It unfortunately seems like it is too late for us to submit another revision now, but the largest edits are that
> >
> > * the title was changed,
> > * the beginning and end of the abstract were changed,
> > * the first paragraph of the introduction was replaced with a new paragraph,
> > * the last paragraph of the introduction (about the difference between the reward hypothesis and the reward-is-enough hypothesis) was cut entirely,
> > * a new Section 1.3 was added, and
> > * the first two paragraphs of the discussion were merged into one new paragraph.
> >
> > There were also some new references added to the 'Related Work'- section, and some minor language edits elsewhere in the paper.

---

### Decision · Program_Chairs · 2023-01-20

**Decision:**

Reject

**Justification For Why Not Higher Score:**

The paper was discussed among all the reviewers, considering the responses and revision. The reviewers shared a common concern that the positioning of the paper has changed after revision and does not read coherently anymore. While there is still a large spread in the reviewers' ratings, the discussions are inclined toward rejection.


**Justification For Why Not Lower Score:**

N/A

**Metareview: Summary, Strengths And Weaknesses:**

The reviewers agreed that the paper provides new insights on the refutation of the reward hypothesis based on three types of tasks that cannot be expressed with scalar rewards. However, the reviewers also raised several concerns and questions in their initial reviews. We thank the authors for their detailed responses and for preparing a revision to address some of these concerns.

There was a large spread in the ratings, and the reviewers extensively discussed this paper, considering the responses and revision. Some of the reviewers raised a common concern that the positioning of the paper has changed after revision (original submission title "The Reward Hypothesis is False"; revised submission title "Three Problem Classes That Markov Rewards Cannot Express"). With this new positioning, the paper does not read coherently and is mostly a list of varied results in three domains. Based on the raised concerns and follow-up discussions, unfortunately, the final decision is a rejection. Nevertheless, this is exciting and potentially impactful work, and we encourage the authors to incorporate the reviewers' feedback when preparing a future revision of the paper.


**Summary Of Ac-Reviewer Meeting:**

Some of the reviewers raised a common concern that the positioning of the paper has changed after revision (original submission title "The Reward Hypothesis is False"; revised submission title "Three Problem Classes That Markov Rewards Cannot Express"). With this new positioning, the paper does not read coherently and is mostly a list of varied results in three different domains. While there is still a large spread in the reviewers' ratings, the discussions are inclined toward rejection.